# Bridging RGB and RAW: Single-step Deterministic Flow with Homogeneous Representation Alignment

**Diedong Feng** [* 1]  **Peiyi Zeng** [* 1]  **Zhen Liu** [1]  **Zhongyang Li** [1]  **Bing Zeng** [1]  **Shuaicheng Liu** [1]

## Abstract

Reconstructing high-fidelity RAW sensor data from processed RGB images is a fundamental yet ill-posed problem, plagued by irreversible information loss and complex non-linear ISP transformations. While generative models offer high-quality reconstruction, they suffer from prohibitive computational costs. Conversely, dominant regression-based methods are fast but susceptible to incoherent observational deviations, often yielding over-smoothed predictions that drift from the authentic signal manifold. To reconcile this trade-off, we propose SHADE, a Single-step Homogeneous Aligned DEterministic flow framework. We validate that, unlike point-to-point regression, the single-step deterministic flow captures global transport trends and enables intrinsic robustness against input perturbations. Furthermore, we introduce Homogeneous Representation Alignment to maximize fidelity. By leveraging a homogeneously initialized student-teacher DINO pair, this mechanism enforces alignment within a shared feature space, significantly amplifying the representational capacity. Extensive experiments demonstrate that SHADE achieves state-of-the-art performance on multiple benchmarks, establishing a new paradigm for accurate and efficient sensor data reconstruction.

## 1. Introduction

In recent years, the computer vision community has shown a surging interest in exploiting RAW sensor data. Compared to processed RGB images, RAW data preserves native radiometric linearity and a wider dynamic range, which simplifies the modeling of physical scene properties and benefits vari-

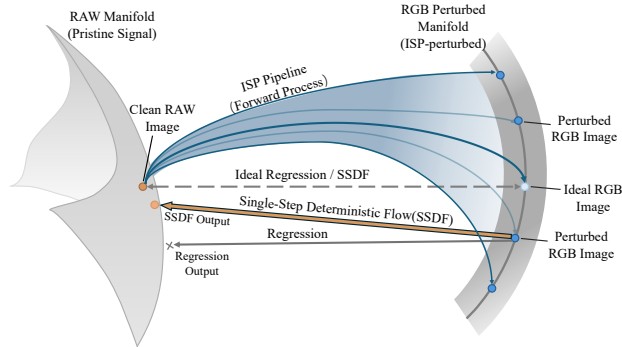

*Figure 1.* **Manifold interpretation of RGB-to-RAW reconstruction.** The forward ISP pipeline maps sensor-level RAW signals onto an ISP-perturbed RGB manifold. Direct regression is sensitive to point-wise deviations and may drift off the RAW manifold, while single-step deterministic flow (SSDF) captures the global transport trend for robust alignment.

ous downstream tasks, such as super resolution (Xu et al., 2019; Conde et al., 2024), image deblurring (Liang et al., 2020; Jiao et al., 2025), low-light enhancement (Chen et al., 2018; Jiang et al., 2025) and object detection (Li et al., 2025; Guo et al., 2025). However, the acquisition of large-scale, diverse RAW datasets is notoriously difficult, and the storage of high-bit-depth sensor data imposes prohibitive overheads. Consequently, RGB-to-RAW reconstruction has emerged as a critical research direction. By effectively recovering sensor-level signals from widely available RGB content, this task offers a promising pathway to democratize RAW-based vision, mitigating the reliance on specialized hardware-dependent data collection.

Despite its practical promise, this inverse problem is inherently ill-posed. The forward ISP pipeline introduces complex non-linear transformations, quantization errors, and irreversible information loss, making the mapping from the degraded RGB observation back to the target RAW manifold highly ambiguous (Brooks et al., 2019). Consequently, recovering the underlying signal structure from such damaged observations remains a major hurdle for modern reconstruction algorithms, especially when fine-grained radiometric cues and high-frequency details are severely distorted.

To address this challenge, deep learning approaches have

---
[*]Equal contribution  [1]University of Electronic Science and Technology of China, China. Correspondence to: Shuaicheng Liu <liushuaicheng@uestc.edu.cn>.

*Proceedings of the 43rd International Conference on Machine Learning*, Seoul, South Korea. PMLR 306, 2026. Copyright 2026 by the author(s).

dominated the field. Currently, the majority of existing industrial solutions model the task as a direct regression problem (Berdan et al., 2025; Zamir et al., 2020; Schwartz et al., 2018). However, regression-based models tend to output the statistical average of all possible solutions when facing the one-to-many nature of inverse problem (Saharia et al., 2022). This not only leads to predictions that drift from the authentic signal manifold but also causes the loss of high-frequency details, failing to preserve the intrinsic characteristics of sensor data. To mitigate these issues, recent generative methods, such as diffusion-based (Dagli, 2023; Reinders et al., 2025) and flow-based models (Xing et al., 2021; Liu et al., 2026), have been proposed. Although they demonstrate impressive reconstruction capabilities, they typically rely on stochastic sampling from uninformative Gaussian noise, necessitating computationally expensive multi-step iterative inference.

Motivated by these limitations, we re-examine the task through the lens of manifold learning. As visually elucidated in Figure 1, in an ideal lossless scenario, the mapping from RGB back to RAW is theoretically reversible. However, real-world ISP pipelines introduce signal loss and quantization, scattering observations onto a Perturbed RGB Manifold that deviates from the ideal latent position. We identify that direct regression, limited by point-to-point fitting, is sensitive to these incoherent observational artifacts, often producing predictions that drift off the target manifold. In contrast, while RAW-Flow (Liu et al., 2026) pioneered the deterministic flow paradigm by explicitly modeling the transport from the RGB latent distribution to the RAW latent distribution, we find that its Single-step Deterministic Flow (SSDF) formulation captures the global transport trend, effectively filtering these perturbations during learning. This capability allows SSDF to approximate the authentic RAW manifold even when learning from corrupted inputs, demonstrating a natural alignment with the structural challenges of this task. This critical divergence is corroborated by our heuristic experiments with noise-induced shape perturbations, which confirm that SSDF consistently recovers the target distribution despite input perturbations.

Building upon this insight, we present SHADE, a single-step deterministic flow framework with Homogeneous Representation Alignment (HRA) for RGB-to-RAW reconstruction. Specifically, SHADE leverages SSDF as a robust baseline to establish a direct mapping from the RGB latent space to the RAW manifold. To further maximize fidelity, we introduce a novel Homogeneous Representation Alignment (HRA) mechanism. In this design, the encoder is instantiated as a trainable student, initialized with the same weights as a frozen pre-trained teacher. This homogeneity ensures that the student and teacher reside in a shared feature space from the outset, eliminating the misalignment often found in heterogeneous distillation. HRA provides stable representation alignment that enhances the representational capacity,

simplifying the learning of the flow trajectory to ensure high-frequency details are accurately recovered.

Our main contributions are summarized as follows:

- We validate Single-step Deterministic Flow (SSDF) as a robust alternative to direct regression specifically for the RGB-to-RAW task. Our heuristic experiments, which simulate the task's inherent perturbations, reveal that while regression is susceptible to incoherent observational deviations and tends to produce predictions that drift from the precise geometric structure, SSDF leverages a global transport trend to distill consistent manifold mappings from corrupted data, establishing a new paradigm for RGB-to-RAW Reconstruction.

- We propose the Homogeneous Representation Alignment (HRA) architecture to maximize the reconstruction fidelity of the single-step flow. By leveraging a homogeneously initialized student-teacher pair, HRA enforces alignment within a homogeneous feature space, amplifying the representational capacity of the flow for single-step inference.

- We achieve state-of-the-art performance on multiple RGB-to-RAW benchmarks with the proposed SHADE framework. Our method not only outperforms existing regression and generative models in reconstruction accuracy but also maintains high computational efficiency, successfully reconciling the trade-off between quality and speed.

## 2. Related Work

**RGB-to-RAW Reconstruction**   RGB-to-RAW reconstruction seeks to recover high-fidelity sensor data from processed RGB images. This task is ill-posed due to the complex non-linear transformations, quantization errors, and irreversible information loss introduced by the ISP pipeline.

Early calibration-based methods relied on controlled multi-exposure captures (Mitsunaga & Nayar, 1999; Chakrabarti et al., 2009; Debevec & Malik, 2023), which restricted their real-world applicability. Learning-based approaches have since emerged, generally categorized into decomposition-based and end-to-end methods. Decomposition-based approaches (Brooks et al., 2019; Li et al., 2024; Conde et al., 2022), invert ISP stages individually or in groups using specific priors. Conversely, end-to-end methods learn direct mappings; notable examples include CycleISP (Zamir et al., 2020), InvISP (Xing et al., 2021), and ReRAW (Berdan et al., 2025). More recently, generative models have pushed the boundaries of reconstruction fidelity. RAW-Diff (Reinders et al., 2025) employs RGB-guided diffusion, while RAW-Flow (Liu et al., 2026) introduces deterministic latent flow matching with cross-scale guidance. However, main-

taining statistical consistency between reconstructed outputs and authentic RAW sensor distributions still poses a critical, unresolved challenge for RGB-to-RAW reconstruction.

**Flow Matching Models**   Flow matching offers an efficient alternative to diffusion-based models (e.g., DDPM (Ho et al., 2020)) by learning supervised vector fields for direct probability transport, thereby avoiding the instability of stochastic sampling, with recent theoretical studies (Zhou & Liu, 2025; Haviv et al., 2025; Feng et al., 2025) further showing its favorable error behavior, geometric generality, and guidance flexibility compared to diffusion-based formulations. The field has evolved from original formulations (Lipman et al., 2022) to rectified variants (Esser et al., 2024) that yield straighter transport trajectories. Recent advancements, such as MeanFlow (Geng et al., 2025) and consistency flow matching (Yang et al., 2024), have further enabled one-step generation capabilities. In the context of RGB-to-RAW recovery, RAW-Flow (Liu et al., 2026) proposes deterministic flow for high-quality reconstruction but remains bound to multi-step solvers, reflecting a broader reliance on numerical solvers in existing flow-based methods and motivating solver-free, end-to-end transport formulations.

**Representation Alignment for Generative Learning** Representation alignment has proven effective in stabilizing generative training by distilling knowledge from pretrained feature encoders. Methods like REPA (Yu et al., 2025) align noisy intermediate states with encoders such as DINOv2 (Oquab et al., 2023) to accelerate convergence. REPA-E (Leng et al., 2025) extends this to end-to-end VAE-diffusion tuning, while iREPA (Singh et al., 2025) refines the objective to prioritize spatial structure over global semantics. However, existing representation alignment strategies are predominantly investigated within iterative diffusion frameworks, with a primary focus on semantic consistency across noisy intermediate states. Their efficacy in single-step RGB-to-RAW reconstruction thus remains underexplored.

## 3. Heuristic Experiment

We design a heuristic Circle-to-Star recovery task to evaluate the robustness of the Deterministic Flow (Liu et al., 2026) in single step (Single-step Deterministic Flow, SSDF) against input perturbations. This setup serves as a simplified analogy for the non-linear degradation in RGB-to-RAW recovery, where we introduce Gaussian noise with increasing variances to the source circle distribution during both training and inference. Both the direct regression baseline and our SSDF are implemented with a simple two-layer MLP to ensure the comparison focuses on the formulation itself.

Empirical results demonstrate that SSDF exhibits superior stability in Fig. 2. As input noise escalates, the regression

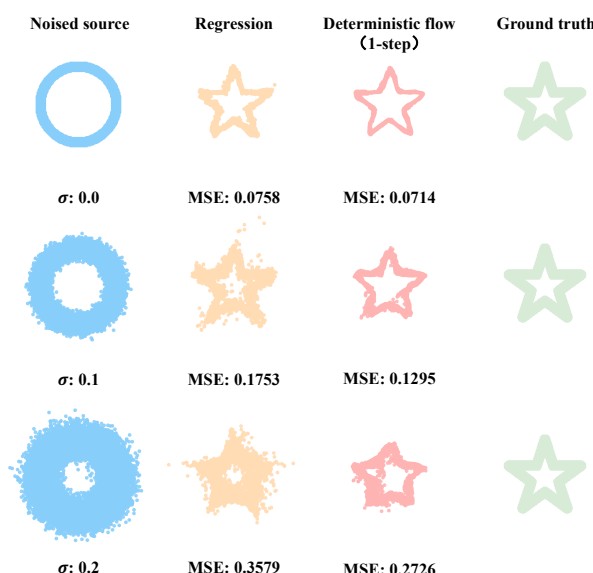

*Figure 2.* **Visual comparison of manifold recovery under different noise levels.** Each row represents the mapping from a source circle to a target star under Gaussian noise $\sigma$. While both methods exhibit some displacement as $\sigma$ increases, the results from SSDF remain structurally intact and significantly more concentrated around the ground truth compared to the scattered predictions of regression. Quantitatively, SSDF maintains a lower MSE across all perturbation levels.

model, limited by point-to-point mapping, produces predictions that significantly drift from the target manifold with higher Mean Squared Error (MSE). In contrast, although all methods exhibit some degree of shift under high noise, SSDF consistently preserves the star geometry with a more compact distribution. This verifies that by learning a global transport trend, the deterministic flow captures a consistent mapping direction from the source to the target distribution. Even in a single step, this paradigm effectively mitigates the influence of incoherent perturbations, validating the core motivation of the SHADE framework.

This difference stems from the fact that SSDF is trained with the deterministic flow-matching objective over the entire linear probability path, learning a time-dependent velocity field that captures the global transport trend instead of independent sample-wise mappings. Its single-step inference therefore arises from the resulting nearly straight and consistent transport trajectory.

## 4. Methodology

### 4.1. Preliminaries

**RGB-to-RAW**   Recovering RAW sensor data from RGB images is an ill-posed inverse problem. The forward Image Signal Processor (ISP) applies non-linear mappings and quantization that cause irreversible information loss (Brooks

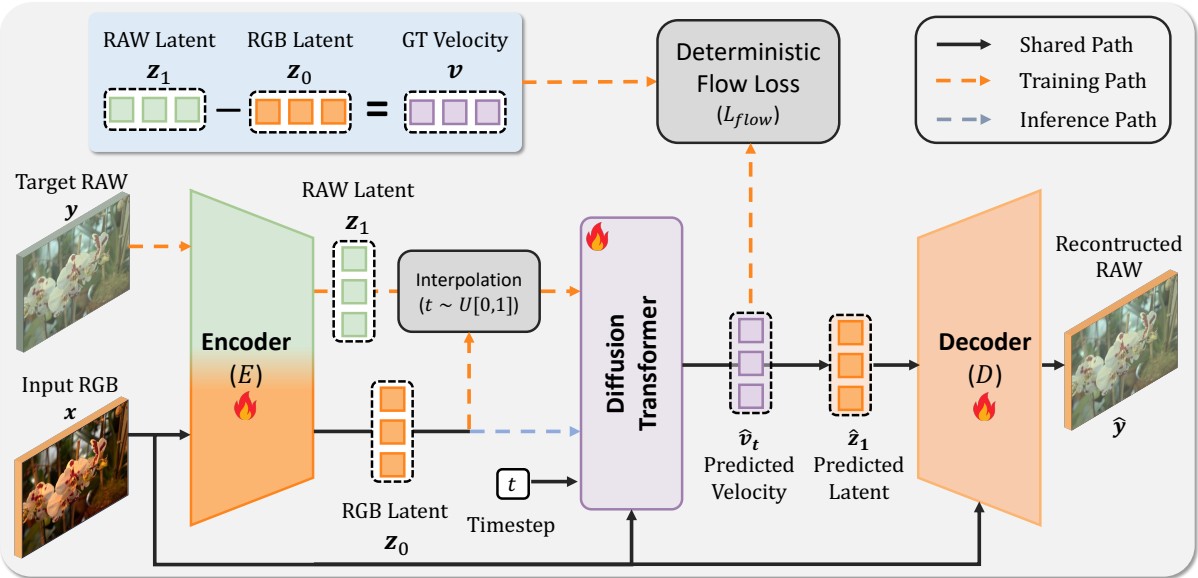

*Figure 3.* **Overall architecture of the proposed SHADE framework.** The training pipeline (orange dashed lines) utilizes a learnable encoder to map input RGB and target RAW images into a shared latent space. A DiT is then trained to predict the deterministic velocity field along a linear probability path. During inference (blue dashed lines), the DiT performs a single-step transport from the RGB latent to the target RAW manifold, which is subsequently reconstructed by the decoder. The framework is optimized through a joint objective of flow matching and image-space reconstruction.

et al., 2019). We model this degradation as

$$\mathbf{x} = \mathcal{P}(\mathbf{y}, \epsilon), \tag{1}$$

where $\mathbf{y} \in \mathcal{Y}$ is the high-fidelity RAW data, $\mathbf{x} \in \mathcal{X}$ is the observed RGB image, $\mathcal{P}$ captures the deterministic non-linearities of the ISP, and $\epsilon$ denotes composite stochastic perturbations from noise and discretization. Under this formulation, the RGB-to-RAW task can be viewed as a manifold recovery problem, where the goal is to learn a mapping $\mathcal{G} : \mathcal{X} \rightarrow \mathcal{Y}$ that projects the perturbed observation $\mathbf{x}$ back onto the original RAW signal manifold.

**Deterministic Flow**  Flow Matching (FM) (Lipman et al., 2022) learns a continuous-time transport from source distribution $p_0(\mathbf{z_0})$ to target distribution $p_1(\mathbf{z_1})$ along a probability path $p_t(\mathbf{z_t})$ governed by the ODE

$$d\mathbf{z_t} = \mathbf{v_t}(\mathbf{z_t} \mid t)\, dt, \quad t \in [0, 1]. \tag{2}$$

The network $v_\theta$ is trained by matching a conditional velocity field to the target path.

In this work, we adopt a deterministic variant based on rectified flow (Liu et al., 2026) with a linear interpolation path $\mathbf{z_t} = (1 - t)\mathbf{z_0} + t\mathbf{z_1}$. The target velocity is constant: $\mathbf{v_t}(\mathbf{z_t}|\mathbf{z_0}, \mathbf{z_1}) = \mathbf{z_1} - \mathbf{z_0}$. The objective simplifies to

$$\mathcal{L}_{\text{flow}}(\theta) = \mathbb{E}_{t \sim \mathcal{U}[0,1]} \big[\|\mathbf{v}_\theta(\mathbf{z_t} \mid t) - (\mathbf{z_1} - \mathbf{z_0})\|_2^2\big], \tag{3}$$

yielding straight trajectories in latent space that support precise single-step deterministic inference.

## 4.2. Overview of SHADE

We formulate the RGB-to-RAW task as a perturbed manifold-to-manifold mapping problem, aiming to learn a deterministic transport from the degraded RGB manifold to the high-fidelity RAW manifold. To this end, we propose SHADE, a single-step framework for efficient latent transport that addresses the non-linear distortions and information loss in the ISP pipeline.

SHADE first maps an input RGB image $\mathbf{x}$ through a learnable encoder $\mathcal{E}$ to obtain the source latent $\mathbf{z}_0$. A Diffusion Transformer (DiT) (Peebles & Xie, 2023) then predicts a vector field $\hat{\mathbf{v}}$ that transports $\mathbf{z}_0$ toward the target RAW manifold. The target latent $\mathbf{z}_1$ is obtained by encoding the ground-truth RAW data with the same $\mathcal{E}$ after reversible dimensional alignment, and the transported latent is decoded back into the RAW domain through a dedicated decoder $\mathcal{D}$. The following sections detail the single-step deterministic flow, the Homogeneous Representation Alignment (HRA) mechanism, and the training strategy.

**Single-step Deterministic Flow**  We construct a deterministic probability path (Liu et al., 2026) $p_t(\mathbf{z}_t)$ connecting the RGB latent $\mathbf{z}_0$ to the RAW latent $\mathbf{z}_1$. During training, we sample $t \sim \mathcal{U}[0, 1]$ and form the intermediate latent via linear interpolation:

$$\mathbf{z}_t = (1 - t)\mathbf{z}_0 + t\mathbf{z}_1. \tag{4}$$

The DiT $\mathbf{v}_\theta$ predicts the conditional vector field $\hat{\mathbf{v}}_t = \mathbf{v}_\theta(\mathbf{z}_t \mid t, \mathbf{x})$, optimized via the flow-matching objective:

$$\mathcal{L}_{\text{flow}}(\theta) = \mathbb{E}_{t \sim \mathcal{U}[0,1]}\left[\|\mathbf{v}_\theta(\mathbf{z_t} \mid t, \mathbf{x}) - (\mathbf{z_1} - \mathbf{z_0})\|_2^2\right]. \quad (5)$$

At inference, we set $t = 0$ to obtain $\hat{\mathbf{v}}$ directly at the starting point, then perform a single Euler step: $\hat{\mathbf{z}}_1 = \mathbf{z}_0 + \hat{\mathbf{v}}$. This single-step procedure avoids multi-step ODE solvers while preserving high-fidelity manifold alignment.

**Homogeneous Representation Alignment**   To strengthen the single-step flow, we introduce Homogeneous Representation Alignment (HRA). The encoder $\mathcal{E}$ is a trainable student encoder initialized from the same pre-trained weights as a frozen pretrained teacher backbone $\mathcal{T}$, specifically instantiated as DINOv3 (Siméoni et al., 2025) in our implementation, ensuring both operate in a shared, homogeneous initial feature space. We adopt DINOv3 to leverage its strong general-purpose representations, and the trainable student encoder further bridges the domain gap between the original pretrained model and RAW-domain images through end-to-end adaptation. The DiT receives the output from $\mathcal{E}$ as input, so its hidden states $\mathbf{h}_t$ naturally align with the pretrained backbone manifold.

We extract the target representation $\mathbf{s}_* = \mathcal{T}(\mathbf{y}) \in \mathbb{R}^{N \times D}$ from the clean RAW data $\mathbf{y}$. A lightweight projection head $g_\phi$ maps DiT hidden states $\mathbf{h}_t$ to $g_\phi(\mathbf{h}_t) \in \mathbb{R}^{N \times D}$. We set the patch size to $16 \times 16$ to maintain consistency with the pre-training configuration of the DINOv3 encoder. The alignment loss maximizes patch-wise cosine similarity after $L_2$ normalization:

$$\mathcal{L}_{\text{align}} = -\mathbb{E}_{t, \mathbf{z}_0, \mathbf{z}_1}\left[\frac{1}{N}\sum_{n=1}^{N}\text{sim}\left(\mathbf{s}_*^{[n]}, g_\phi(\mathbf{h}_t^{[n]})\right)\right]. \quad (6)$$

This homogeneous regularization provides stable representation alignment, enabling the DiT to extract robust features from interpolated latents $\mathbf{z}_t$ and substantially improving single-step reconstruction fidelity.

### 4.3. Training Strategy

The model is trained end-to-end. The complete objective combines three complementary loss terms:

$$\mathcal{L} = \lambda_{\text{flow}}\mathcal{L}_{\text{flow}} + \lambda_{\text{align}}\mathcal{L}_{\text{align}} + \lambda_{\text{rec}}\left(\mathcal{L}_{\text{rec}}^{\text{AE}} + \mathcal{L}_{\text{rec}}^{\text{DiT}}\right), \quad (7)$$

where $\mathcal{L}_{\text{flow}}$ is the flow matching loss (Lipman et al., 2022), $\mathcal{L}_{\text{align}}$ is the homogeneous alignment loss mentioned in Sec. 4.2, and the reconstruction term consists of two image-space supervision signals.

The autoencoder reconstruction loss $\mathcal{L}_{\text{rec}}^{\text{AE}}$ is computed by encoding the ground-truth RAW image $\mathbf{y}$ into its latent $\mathbf{z}_1 = \mathcal{E}(\mathbf{y})$ and decoding it conditioned on the input RGB image $\mathbf{x}$, i.e.,

$$\hat{\mathbf{y}}^{\text{AE}} = \mathcal{D}(\mathbf{z}_1 \mid \mathbf{x}), \quad (8)$$

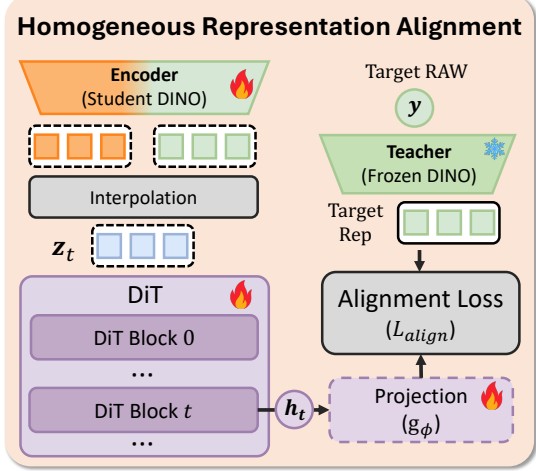

*Figure 4.* **Structure of the Homogeneous Representation Alignment module.** SHADE utilizes a student-teacher DINO (Siméoni et al., 2025) configuration with shared pre-trained weights to establish a homogeneous feature space. HRA aligns DiT hidden states with teacher representations to provide a training-time alignment signal, which facilitates the learning of the flow trajectory for accurate single-step RAW reconstruction.

with the loss measured between $\hat{\mathbf{y}}^{\text{AE}}$ and $\mathbf{y}$. This term ensures the decoder can faithfully recover the RAW image from its own latent representation.

The DiT-guided reconstruction loss $\mathcal{L}_{\text{rec}}^{\text{DiT}}$ uses the DiT-predicted target latent $\hat{\mathbf{z}}_1$, obtained at a randomly sampled $t \sim \mathcal{U}(0, 1)$ as

$$\hat{\mathbf{z}}_1 = \mathbf{z}_t + \mathbf{v}_\theta(\mathbf{z}_t \mid t, \mathbf{x}) \cdot (1 - t), \quad (9)$$

where $\mathbf{z}_t$ is the linearly interpolated latent at time $t$. This latent is decoded under the same conditioning $\mathbf{x}$, i.e.,

$$\hat{\mathbf{y}}^{\text{DiT}} = \mathcal{D}(\hat{\mathbf{z}}_1 \mid \mathbf{x}), \quad (10)$$

and the loss is measured between $\hat{\mathbf{y}}^{\text{DiT}}$ and $\mathbf{y}$. This term ensures the DiT-predicted latent yields a high-fidelity RAW reconstruction when decoded.

Both reconstruction losses are computed as

$$\mathcal{L}_{\text{rec}}(\hat{\mathbf{y}}, \mathbf{y}) = \|\hat{\mathbf{y}} - \mathbf{y}\|_2^2 + \alpha \cdot \mathcal{L}_{\text{logL1}}(\hat{\mathbf{y}}, \mathbf{y}), \quad (11)$$

where the mean squared error promotes pixel-level accuracy and the log-L1 (Reinders et al., 2025) term

$$\mathcal{L}_{\text{logL1}}(\hat{\mathbf{y}}, \mathbf{y}) = \frac{1}{CHW}\|\log(\hat{\mathbf{y}} + \epsilon) - \log(\mathbf{y} + \epsilon)\|_1 \quad (12)$$

preserves relative intensity relationships in the RAW domain, where we set $\epsilon = 2^{-14}$ for numerical stability, and set $\alpha = 0.1$ as the weight of Log-L1 term.

*Table 1.* Quantitative comparisons with state-of-the-art methods on various datasets. **Bold**: best, underline: second best.

| Method | Reference | PASCALRAW | | FiveK-Nikon | | FiveK-Canon | | NOD-Nikon | | NOD-Sony | |
|---|---|---|---|---|---|---|---|---|---|---|---|
| | | PSNR↑ | SSIM↑ | PSNR↑ | SSIM↑ | PSNR↑ | SSIM↑ | PSNR↑ | SSIM↑ | PSNR↑ | SSIM↑ |
| UNet | MICCAI'15 | 33.87 | 0.9645 | 25.65 | 0.8535 | 29.20 | 0.9221 | 37.52 | 0.9708 | 36.47 | 0.9545 |
| UPI | CVPR'19 | 26.26 | 0.8828 | 25.55 | 0.8416 | 27.24 | 0.8848 | 25.47 | 0.4803 | 24.73 | 0.4363 |
| CycleISP | CVPR'20 | 34.89 | 0.9735 | 26.41 | 0.8574 | 29.96 | 0.9389 | – | – | – | – |
| DDPM | NeurIPS'20 | 32.24 | 0.9574 | 25.82 | 0.8392 | 28.70 | 0.8485 | 36.41 | 0.9554 | 33.90 | 0.9189 |
| InvISP | CVPR'21 | 28.47 | 0.8631 | 26.94 | 0.8268 | 28.32 | 0.8648 | 36.91 | 0.9108 | 37.29 | 0.9379 |
| CycleR2R | TPAMI'24 | 26.19 | 0.8349 | 25.43 | 0.8272 | 26.33 | 0.8579 | 27.33 | 0.6144 | 22.50 | 0.4629 |
| RAW-Diff | WACV'25 | 29.54 | 0.9340 | 26.96 | 0.8608 | 31.84 | **0.9571** | 39.00 | 0.9608 | 37.08 | 0.9252 |
| ReRAW-R | CVPR'25 | 35.96 | 0.9834 | 27.90 | 0.8447 | 30.98 | 0.9230 | 38.43 | 0.9758 | 36.50 | **0.9813** |
| ReRAW-S | CVPR'25 | 37.29 | **0.9854** | 28.04 | 0.8377 | 31.76 | 0.9328 | 39.11 | 0.9753 | 38.26 | 0.9782 |
| RAW-Flow | AAAI'26 | 37.62 | 0.9831 | 30.79 | 0.8772 | 32.55 | 0.9445 | – | – | – | – |
| Ours | – | **39.07** | 0.9849 | **30.89** | **0.8858** | **33.69** | 0.9495 | **40.61** | **0.9856** | **39.07** | 0.9687 |

# 5. Experiments

## 5.1. Experimental Settings

**Datasets** We assess the performance of our proposed approach on five distinct RAW image datasets acquired from a variety of digital single-lens reflex camera (DSLR) and mirrorless camera models. Drawing from the MIT-Adobe FiveK dataset (Bychkovsky et al., 2011), we adhere to the methodology in RAW-Diff (Reinders et al., 2025) to create the FiveK-Nikon and FiveK-Canon subsets, comprising 590 RAW images captured with a Nikon D700 and 777 RAW images from a Canon EOS 5D, respectively. Additionally, we incorporate the PASCALRAW dataset (Omid-Zohoor et al., 2015), gathered using a Nikon D3200. To evaluate resilience in challenging low-light scenarios, we derive two additional subsets from the NOD nighttime RAW dataset (Morawski et al., 2022), specifically NOD-Nikon and NOD-Sony, each consisting of 600 RAW images obtained via a Nikon D750 and a Sony RX100 VII. Across all datasets, we apply a random partition of 85% for training and 15% for testing. The associated sRGB images are produced from the RAW files utilizing the RawPy library, yielding spatially registered, high-resolution sRGB-RAW pairs suitable for model training and assessment.

**Implementation Details** Our method is implemented using the PyTorch framework and trained with the Adam (Kingma & Ba, 2014) optimizer, with hyperparameters set to $\beta_1 = 0.9$, $\beta_2 = 0.999$. The initial learning rate is set to $5 \times 10^{-4}$ across all network components, coupled with a Reduce-On-Plateau scheduler to enable adaptive learning rate reduction. Given that the framework supports end-to-end optimization without requiring separate stages for distinct modules, training is conducted in a unified manner over 400 epochs. All experiments are performed on NVIDIA RTX 4090D GPUs.

**Evaluation Metrics** In line with established practices in the field, we utilize Peak Signal-to-Noise Ratio (PSNR) and Structural Similarity Index Measure (SSIM) (Wang et al., 2004) to quantify the fidelity of the reconstructed RAW images. These metrics are calculated directly on the full-resolution outputs in the RAW domain, with all presented results representing averages computed across the reconstructed samples in each respective test partition.

## 5.2. Comparison with Existing Methods

**Compared Methods** To comprehensively evaluate the performance of the proposed SHADE framework, we compare it with various state-of-the-art RGB-to-RAW reconstruction methods, including UNet (Ronneberger et al., 2015), UPI (Brooks et al., 2019), CycleISP (Zamir et al., 2020), DDPM (Ho et al., 2020), InvISP (Xing et al., 2021), CycleR2R (Li et al., 2024), RAW-Diff (Reinders et al., 2025), ReRAW-R and ReRAW-S (Berdan et al., 2025), and RAW-Flow (Liu et al., 2026). For a fair comparison, all methods are evaluated using their official implementations and default settings.

**Quantitative Comparison** Table 1 presents quantitative comparisons between our SHADE method and various state-of-the-art approaches on five benchmark datasets for RGB-to-RAW recovery. Our method consistently achieves state-of-the-art performance, securing the highest PSNR scores across all datasets and leading SSIM metrics on most. For example, on the challenging NOD-Nikon dataset, SHADE attains a PSNR of 40.61 dB, outperforming the second-best method by 1.50 dB in PSNR, which highlights its exceptional ability to handle complex degradations with high fidelity. Compared to regression-based methods like UNet, UPI, and ReRAW variants, SHADE demonstrates substantial advantages in reconstruction quality, with PSNR improvements surpassing many existing methods. Generative baselines such as DDPM, RAW-Diff, and RAW-Flow, while competitive in isolated metrics, fall short in overall balance, often lagging by over 1 dB in PSNR due to their reliance on multi-step sampling or noise initialization. These results un-

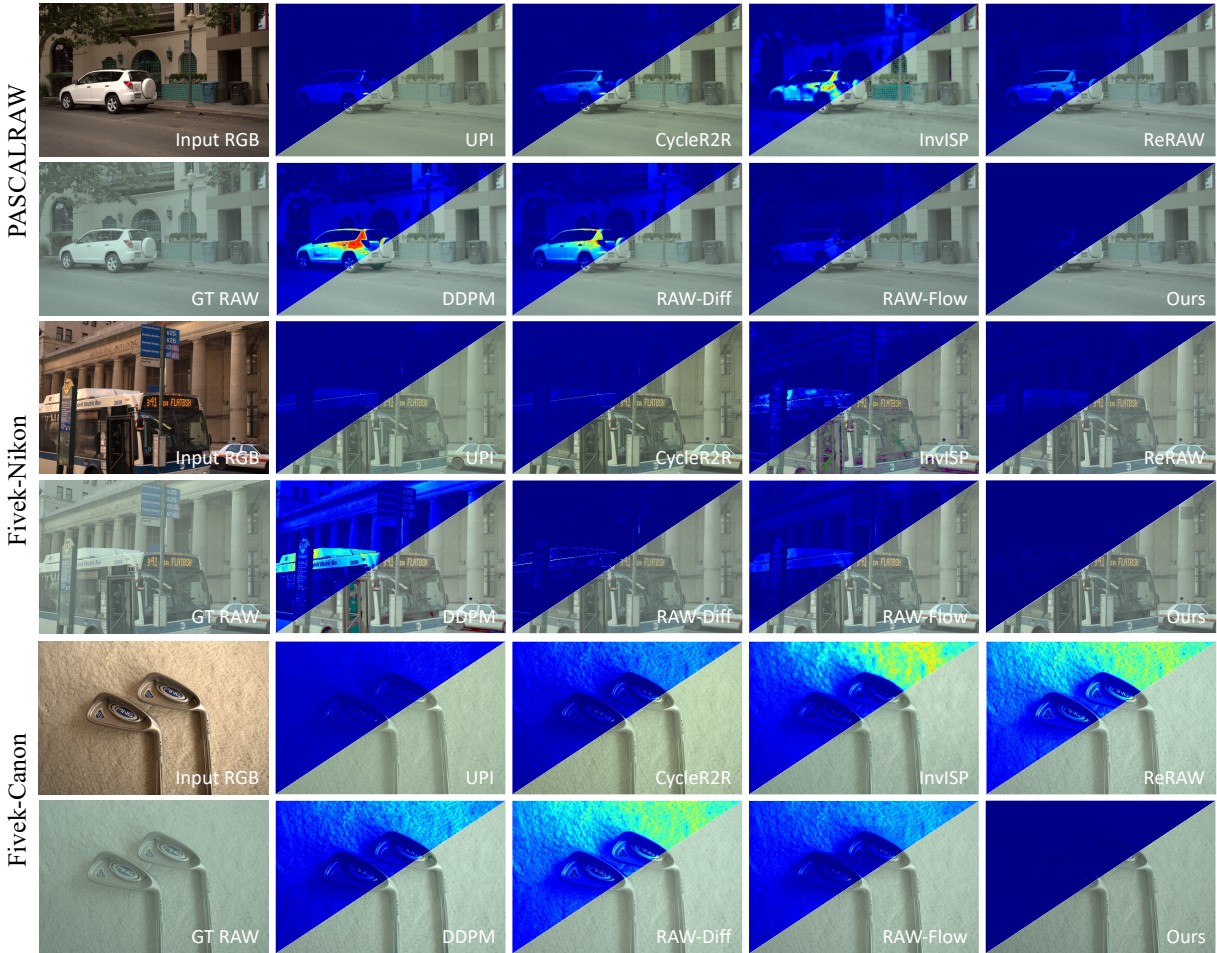

*Figure 5.* **Qualitative comparison on daylight datasets.** For each method, the top-left inset shows the reconstruction error map (darker regions indicate smaller errors), and the bottom-right inset shows the reconstructed RAW visualization. Compared with existing regression and generative methods, our SHADE effectively restores sharp structural details and preserves color consistency in regions corrupted by ISP-induced quantization and non-linear distortion.

derscore SHADE's effectiveness in combining deterministic flow efficiency with robust generalization.

**Qualitative Comparison** Fig. 5 and Fig. 6 showcase the qualitative results on three daylight datasets (i.e., PASCAL-RAW, FiveK-Nikon, and FiveK-Canon) and two nighttime benchmarks (i.e., NOD-Nikon and NOD-Sony), respectively. In daylight scenes, prior methods like UPI (Brooks et al., 2019), CycleISP (Zamir et al., 2020), CycleR2R (Li et al., 2024), and ReRAW (Berdan et al., 2025) suffer from color deviations and detail loss due to accumulated errors from non-linear ISP inversions, especially around fine structures and object boundaries. While generative methods such as RAW-Diff (Reinders et al., 2025) and RAW-Flow (Liu et al., 2026) produce clean reconstructions, they struggle to maintain global illumination consistency and may introduce slight brightness shifts. In contrast, SHADE, enhanced by the DINO-distilled priors from the representation alignment

of HRA, achieves stronger color and structural fidelity, effectively restoring details in regions affected by quantization and non-linear distortions.

On nighttime datasets, competing methods are often disturbed by severe low-light noise and clipped saturated regions, leading to over-smoothed or artifact-ridden outputs. These issues are more evident near saturated light sources and underexposed backgrounds, where the RGB observation contains limited radiometric information. In contrast, SHADE exhibits strong robustness against such perturbations: its deterministic flow captures the global transport trend to suppress noise-induced deviations, while the aligned representation prior further improves manifold alignment under severe information loss. As a result, our method better recovers clipped highlights and maintains more consistent color responses across low-light scenes. These visual improvements are consistent with the quantita-

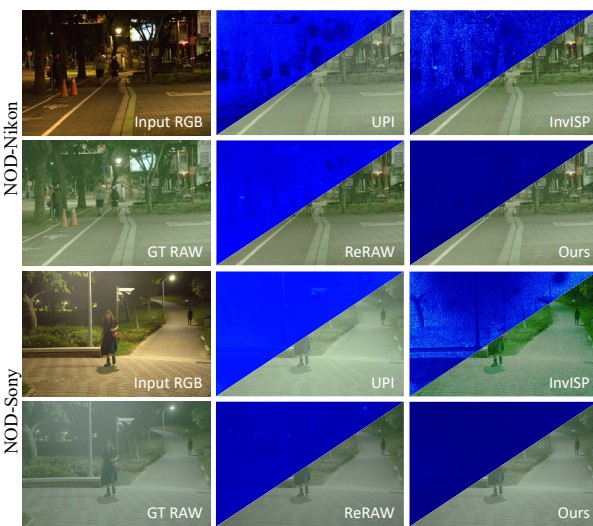

Figure 6. **Qualitative comparison on nighttime datasets.** In extreme low-light and saturated scenarios, our method faithfully reconstructs clipped signals and authentic sensor noise, whereas other methods often suffer from over-smoothing or severe artifacts.

tive gains, further demonstrating the robustness of SHADE under challenging illumination conditions.

### 5.3. Ablation Studies

**Single-step Deterministic Flow** We compare our Single-step Deterministic Flow (SSDF) against multi-step generative baselines (standard Flow Matching (Lipman et al., 2022) and DDIM (Song et al., 2021) with 10 steps), other single-step methods starting from noise (Mean Flow (Geng et al., 2025) and Consistency Flow Matching (Yang et al., 2024)), and a direct regression model. For each baseline, we report its best-practice setting and enable HRA to ensure a fair comparison under near-optimal configurations. As shown in Table 2, SSDF achieves the highest performance while requiring only one inference step. Multi-step methods, despite iterative refinement, are outperformed, and noise-initiated single-step approaches also yield lower results. Compared with direct regression, which suffers from sensitivity to pathological perturbations, SSDF captures the global transport trend, effectively filtering these perturbations during learning, thereby delivering higher reconstruction fidelity under complex degradations. Thus, SSDF attains multi-step-level quality at single-step speed, demonstrating a compelling accuracy-efficiency advantage for RGB-to-RAW recovery.

**Homogeneous Representation Alignment** We conducted an ablation study to assess the contributions of weight loading and representation alignment in the proposed Homogeneous Representation Alignment (HRA). Four configurations were compared, and the results are summarized

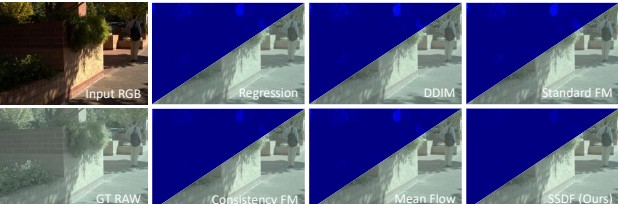

Figure 7. Visual comparisons of our Single-step Deterministic Flow (SSDF) against multi-step generative baselines, other single-step methods, and a direct regression model.

Table 2. Quantitative comparison on FiveK-Nikon and PASCALRAW datasets. We compare SSDF against Standard Flow Matching (FM), Consistency FM, and other baselines. Our SSDF achieves the best performance with a single inference step.

| Variant | FiveK-Nikon | | PASCALRAW | |
|---|---|---|---|---|
| | PSNR ↑ | SSIM ↑ | PSNR ↑ | SSIM ↑ |
| DDIM | 30.78 | 0.8856 | 37.53 | 0.9850 |
| FM | 30.45 | 0.8696 | 37.35 | 0.9830 |
| Mean Flow | 30.38 | 0.8729 | 37.97 | 0.9815 |
| Consistency FM | 30.29 | 0.8757 | 37.80 | 0.9844 |
| Regression | 30.32 | 0.8742 | 37.69 | **0.9851** |
| SSDF (Ours) | **30.89** | **0.8858** | **39.07** | 0.9850 |

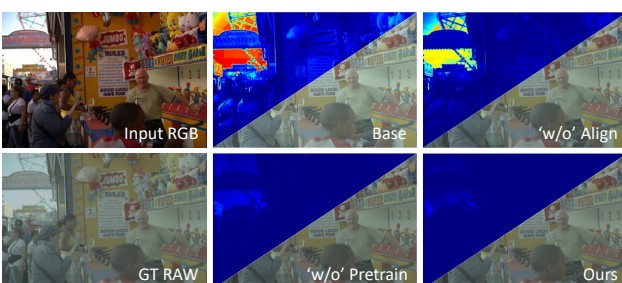

Figure 8. Visual comparisons of our ablation study on the weight loading and representation alignment components of the proposed Homogeneous Representation Alignment (HRA).

in Table 3. The baseline without weight loading or alignment achieves the lowest performance. Initializing the student DINO encoder with the teacher's weights brings moderate gains by providing a consistent and meaningful starting latent space. Applying representation alignment alone also improves the results by enforcing representation consistency between the DiT intermediate features and the teacher. The combination of both weight loading and representation alignment yields the highest performance, outperforming all other configurations. These results demonstrate that the two components are complementary: pre-aligned initialization establishes a strong starting point, while representation alignment effectively refines the latent representations during training, leading to superior final recovery quality.

**Loss Terms** We further conduct ablation experiments on $L_2$, log-$L_1$, $\mathcal{L}_{\text{rec}}^{\text{AE}}$, and $\mathcal{L}_{\text{rec}}^{\text{DiT}}$, as summarized in Table 4. These

*Table 3.* Quantitative ablation study of HRA components on FiveK-Nikon and PASCALRAW datasets. ↑ indicates higher is better. The combination of weight loading and representation alignment yields the highest performance, demonstrating their complementary roles.

| Variant | FiveK-Nikon | | PASCALRAW | |
|---|---|---|---|---|
| | PSNR ↑ | SSIM ↑ | PSNR ↑ | SSIM ↑ |
| Base | 28.05 | 0.8418 | 37.19 | 0.9848 |
| 'w/o' Pretrain | 30.01 | 0.8621 | 38.29 | **0.9873** |
| 'w/o' Align | 29.39 | 0.8472 | 38.60 | 0.9828 |
| Ours | **30.89** | **0.8858** | **39.07** | 0.9850 |

*Table 4.* Quantitative ablation study of individual loss terms on FiveK-Nikon and PASCALRAW datasets.

| Variant | FiveK-Nikon | | PASCALRAW | |
|---|---|---|---|---|
| | PSNR ↑ | SSIM ↑ | PSNR ↑ | SSIM ↑ |
| w/o $L_2$ | 29.62 | 0.8713 | 37.59 | **0.9853** |
| w/o log-$L_1$ | 30.32 | **0.8879** | 35.51 | 0.9667 |
| w/o $\mathcal{L}_{rec}^{AE}$ | 29.67 | 0.8761 | 36.83 | 0.9819 |
| w/o $\mathcal{L}_{rec}^{DiT}$ | 29.21 | 0.8736 | 38.16 | 0.9844 |
| Full (Ours) | **30.89** | 0.8858 | **39.07** | 0.9849 |

loss terms supervise different aspects of the reconstruction process. The $L_2$ loss encourages absolute radiance fidelity and is especially useful for preserving high-intensity responses, while the log-$L_1$ loss emphasizes relative intensity consistency and improves the recovery of low-light details. The autoencoding reconstruction loss $\mathcal{L}_{rec}^{AE}$ ensures that the decoder maintains a stable mapping from the RAW latent space back to image space, providing a reliable reconstruction basis. In contrast, $\mathcal{L}_{rec}^{DiT}$ exposes the decoder to DiT-predicted latents and reduces the mismatch between training-time autoencoded latents and inference-time transported latents. As shown in Table 4, removing any single term generally degrades the overall performance, indicating that these objectives are not redundant. In particular, applying the DiT reconstruction constraint on top of the AE constraint improves the final output, whereas relying on it alone can weaken the decoder's intrinsic reconstruction ability and lead to inferior results. These results confirm that the pixel-level, logarithmic, autoencoding, and DiT-guided objectives jointly provide complementary supervision for robust RGB-to-RAW reconstruction.

## 6. Conclusion

In this paper, we presented SHADE, a single-step deterministic flow framework for high-fidelity RGB-to-RAW reconstruction. By integrating Single-step Deterministic Flow (SSDF) with Homogeneous Representation Alignment (HRA), SHADE mitigates ISP-induced perturbations and achieves precise latent alignment in a single inference step. Experiments across five benchmarks demonstrate that

our approach outperforms state-of-the-art regression and generative methods in restoring high-frequency details and faithful RAW signal characteristics.

**Limitations and Future Work.** Despite its effectiveness, SHADE still has several limitations. First, as a deterministic transport model, SHADE prioritizes accurate recovery of ISP nonlinearities and HDR signals, but it cannot explicitly sample diverse RAW noise realizations from the same RGB observation. Second, our current evaluation follows the camera-specific paired-data protocol adopted by recent RGB-to-RAW methods, where training and testing pairs are generated from the same camera RAW files. Exploring cross-sensor RGB-to-RAW reconstruction is a promising direction, and we will investigate multi-sensor training and cross-ISP adaptation in future work.

## Impact Statement

This work aims to advance low-level vision by reconstructing RAW sensor data from RGB images and by providing higher-quality synthetic sensor data for downstream applications, such as object detection, low-light enhancement, and scientific imaging. These capabilities can benefit the computer vision community by reducing reliance on specialized RAW data collection and improving data availability for sensor-aware vision tasks. However, the ability to generate plausible RAW-like images from RGB inputs may also introduce potential negative societal impacts. In particular, such techniques could be misused to forge RAW-like evidence or to bypass forensic and authenticity checks that rely on the assumed trustworthiness of RAW data. We therefore emphasize that reconstructed RAW images should not be treated as original sensor measurements in forensic or evidentiary contexts, and future deployment should consider appropriate provenance tracking, disclosure, and authenticity verification mechanisms.

## Acknowledgements

This work was supported in part by the National Natural Science Foundation of China (NSFC) under grant 62372091, and in part by the Hainan Province Science and Technology Plan Project under Grant ZDYF2024(LALH)001.

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

# A. Network Architecture and Implementation Details

## A.1. Detailed Network Architecture

The proposed SHADE framework consists of a Student-Teacher Encoder pair and a flow-estimating Diffusion Transformer (DiT). Below, we detail the specific configurations of each component.

**Student-Teacher Encoders**  To enable Homogeneous Representation Alignment (HRA), we instantiate both the student and teacher encoders using the DINOv3 (Siméoni et al., 2025) architecture. Both encoders map the input RGB images into a compact latent space with a feature dimension of $D_{enc} = 384$. The teacher encoder is frozen during training, while the student encoder is initialized with the teacher's weights and remains trainable to adapt to the specific domain.

**Diffusion Transformer (DiT)**  Our flow estimator is based on a Diffusion Transformer architecture, designed to predict the vector field for the single-step transport. The architecture consists of a lightweight Transformer backbone followed by a DiT head for feature adaptation:

- **Deep Backbone:** The core processing unit is a Transformer backbone consisting of **14 layers** with a hidden dimension of **128**. This lightweight design ensures computational efficiency while maintaining sufficient capacity for flow estimation.

- **DiT Head:** After the backbone, the hidden features are processed by a **2-layer** head with a hidden dimension of **256**. This head increases the feature dimension and adapts the DiT representations to the high-dimensional DINO feature space used by HRA.

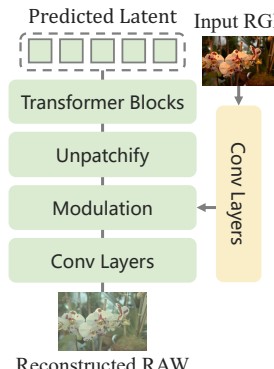

*Figure S1.* **Architecture of the decoder.**

**Decoder**  The decoder architecture is illustrated in Figure S1. Multiple Transformer blocks progressively reconstruct the RAW output from the transported latent, while the RGB observation participates in feature modulation to help preserve scene structure and recover RAW-domain intensity details.

**HRA Projection and Alignment**  To enforce the representation alignment constraint, we introduce a projection mechanism to bridge the dimension of the DiT and the DINOv3 encoders.

- **Feature Extraction:** We extract the intermediate hidden states from the **7th layer** of the DiT backbone.

- **Projection Layer:** These extracted features are fed into a projection layer that maps the DiT's hidden dimension (128) to the target alignment dimension of **384**, matching the output space of the DINOv3 encoders.

## A.2. Hyperparameter Settings

**Loss Weights.**  The total training objective is a weighted sum of the flow matching loss ($\mathcal{L}_{flow}$), the reconstruction loss ($\mathcal{L}_{rec}$), and the homogeneous aligned guidance loss ($\mathcal{L}_{align}$). In our experiments, we assign equal importance to all components:

$$\mathcal{L} = \lambda_{flow}\mathcal{L}_{flow} + \lambda_{align}\mathcal{L}_{align} + \lambda_{rec}\left(\mathcal{L}_{rec}^{AE} + \mathcal{L}_{rec}^{DiT}\right), \tag{13}$$

where we set $\lambda_{flow} = 1$, $\lambda_{rec} = 1$, and $\lambda_{align} = 1$.

**Training Configurations.**  Table S1 summarizes the detailed hyperparameters used in our implementation.

# B. Reversible Dimensional Alignment

To bridge the dimensional gap between the packed RAW input ($\frac{H}{2} \times \frac{W}{2} \times 4$) and the standard RGB feature space ($H \times W \times 3$) without information loss, we employ a Reversible Dimensional Alignment (RDA) module. Unlike traditional demosaicing methods that introduce interpolation artifacts, RDA performs a deterministic pixel rearrangement that is theoretically invertible.

*Table S1.* Detailed hyperparameter settings for the SHADE framework.

| Hyperparameter | Value |
| --- | :---: |
| *Architecture Specifications* | |
| Encoder Model | DINOv3 |
| Encoder Dimension ($D_{\text{enc}}$) | 384 |
| DiT Backbone Layers | 14 |
| DiT Backbone Hidden Size | 128 |
| DiT Head Layers | 2 |
| DiT Head Hidden Size | 256 |
| HRA Extraction Layer | $7^{\text{th}}$ Layer |
| *Optimization* | |
| Optimizer | Adam |
| Learning Rate | $5 \times 10^{-4}$ |
| Betas ($\beta_1, \beta_2$) | $(0.9, 0.999)$ |
| Weight Decay | 0 |
| Epsilon for Adam ($\epsilon$) | $1 \times 10^{-8}$ |
| Batch Size | 8 |
| Total Epochs | 400 |
| Loss Weights ($\lambda_{\text{flow}}, \lambda_{\text{rec}}, \lambda_{\text{align}}$) | $1, 1, 1$ |

## B.1. Formal Formulation

Let $\mathbf{X} \in \mathbb{R}^{B \times 4 \times \frac{H}{2} \times \frac{W}{2}}$ denote the input packed RAW tensor, where the four channels correspond to the Bayer pattern components $\{R, G_1, G_2, B\}$. The target aligned tensor is denoted as $\mathbf{Y} \in \mathbb{R}^{B \times 3 \times H \times W}$, matching the spatial resolution of the standard RGB image. The forward alignment function $\mathcal{F}_{\text{align}} : \mathbb{R}^{4 \times \frac{H}{2} \times \frac{W}{2}} \to \mathbb{R}^{3 \times H \times W}$ maps each pixel $(h, w)$ in the source domain (where $h < \frac{H}{2}, w < \frac{W}{2}$) to a $2 \times 2$ block in the target domain according to the Bayer geometry.

The anchor values in $\mathbf{Y}$ are populated as follows:

$$\mathbf{Y}_{0,2h,2w} = \mathbf{X}_{0,h,w} \quad \text{(Red)} \tag{14}$$

$$\mathbf{Y}_{1,2h,2w+1} = \mathbf{X}_{1,h,w} \quad \text{(Green}_1\text{)} \tag{15}$$

$$\mathbf{Y}_{1,2h+1,2w} = \mathbf{X}_{2,h,w} \quad \text{(Green}_2\text{)} \tag{16}$$

$$\mathbf{Y}_{2,2h+1,2w+1} = \mathbf{X}_{3,h,w} \quad \text{(Blue)} \tag{17}$$

where subscripts denote (channel, height, width) indices. The remaining spatial positions in $\mathbf{Y}$ are filled via a nearest-neighbor-based interpolation $\Phi(\cdot)$ to ensure spatial continuity for feature extraction, formulated as:

$$\mathbf{Y} \leftarrow \Phi(\mathbf{Y}_{\text{anchors}}). \tag{18}$$

Crucially, the inverse operation $\mathcal{F}_{\text{align}}^{-1}$ simply samples the values at the anchor indices defined in Eq. (1)-(4), discarding the interpolated values, thus guaranteeing perfect reconstruction: $\mathbf{X} = \mathcal{F}_{\text{align}}^{-1}(\mathcal{F}_{\text{align}}(\mathbf{X}))$.

## C. Computational Efficiency Analysis

To assess the practical feasibility of SHADE for real-world deployment, particularly in high-resolution photography, we benchmark the inference latency against both generative and regression-based methods. Table S2 details the inference time (in seconds) across four resolution scales ranging from $512 \times 512$ to $4096 \times 4096$. We also report the parameter count and FLOPs at $4096 \times 4096$ resolution to provide a more complete comparison of model complexity. All measurements were conducted on a single NVIDIA RTX 4090 GPU.

### C.1. Comparison with Generative Methods

Standard generative baselines suffer from high latency due to their iterative sampling requirements. For instance, at $4096 \times 4096$ resolution, DDIM (Song et al., 2021) (20 steps) requires 35.12 seconds, and RAW-Flow (Liu et al., 2026) (20 steps) requires 9.35 seconds. In contrast, by reducing the transport process to a single step, SHADE completes the same task in only 1.51 seconds. This represents a speedup of approximately $23\times$ compared to DDIM and $6\times$ compared

to RAW-Flow, significantly narrowing the gap between generative quality and real-time processing. In terms of model complexity, SHADE also achieves the second lowest parameter count and FLOPs among the generative methods, indicating that its speed advantage does not rely on excessive model capacity or computation.

## C.2. Comparison with Regression Methods

When compared to direct regression methods, SHADE demonstrates competitive efficiency with superior scalability. While ReRAW (Berdan et al., 2025) achieves marginally lower latency at lower resolutions ($512 \times 512$ and $1024 \times 1024$), SHADE surpasses it as the resolution increases. At $2048 \times 2048$ and $4096 \times 4096$, SHADE achieves the lowest inference times among all compared methods (0.39s and 1.51s, respectively). This indicates that our architecture and the single-step flow formulation scale more effectively to the high-megapixel demands of modern imaging sensors.

*Table S2.* Inference time (in seconds) of different methods at various image resolutions. Params and FLOPs are measured at $4096 \times 4096$ resolution. SHADE achieves the best performance at high resolutions while maintaining a significant speed advantage over iterative generative baselines.

| Method | 512×512 | 1024×1024 | 2048×2048 | 4096×4096 | Params (M) | FLOPs (T) |
|---|---|---|---|---|---|---|
| DDIM (Song et al., 2021) (20 steps) | 0.3494 | 2.2527 | 8.7687 | 35.1210 | 58.06 | 764.75 |
| RAW-Diff (Reinders et al., 2025) (24 steps) | 1.7933 | 2.3613 | 4.3903 | 13.7905 | 25.00 | 1179.92 |
| RAW-Flow (Liu et al., 2026) (20 steps) | 0.2209 | 1.1273 | 2.9507 | 9.3551 | 72.19 | 135.84 |
| CycleISP (Zamir et al., 2020) | 0.0677 | 0.2807 | 1.2040 | 5.2897 | **3.14** | 117.33 |
| ReRAW (Berdan et al., 2025) | **0.0275** | **0.0986** | 0.4380 | 2.0502 | 23.77 | **20.96** |
| SHADE (Ours) | 0.0452 | 0.1220 | **0.3940** | **1.5117** | 57.09 | 212.76 |

# D. Analysis of Flow Trajectory Linearity

To theoretically justify the single-step inference strategy employed in our framework, we investigate the geometric properties of the learned transport dynamics. Specifically, we analyze the trajectory linearity of the underlying Deterministic Flow. The core premise is that if the learned vector field transports samples along a straight path, the instantaneous velocity direction should remain constant throughout the integration process. Therefore, determining the straightness of the flow is equivalent to evaluating the directional consistency of velocity vectors across different time steps.

## D.1. Evaluation Methodology

We perform numerical integration on the PASCALRAW (Omid-Zohoor et al., 2015) validation set using a multi-step solver to simulate the full transport trajectory. During this process, we record the predicted instantaneous velocity vector $v_t$ at each discretized time step $t \in [0, 1]$. To quantify linearity, we calculate the cosine similarity between the velocity vectors at different temporal locations. High similarity scores indicate that the flow direction is stable and time-invariant, corresponding to a straight trajectory.

This analysis validates the rationale behind our Single-step Deterministic Flow (SSDF). While standard flow matching formulations may induce curved trajectories requiring multi-step integration, our SSDF is a specialized configuration where the model is successfully optimized to learn a rectified, straight-line flow. This geometric characteristic allows us to reduce the inference steps to $N = 1$ without sacrificing reconstruction fidelity.

## D.2. Visual Comparison

Figure S2 visualizes the velocity consistency of a standard Flow Matching (FM) baseline compared to our method. The plots illustrate the directional correlation of the velocity field over the integration time.

As shown in Figure S2 (a), our method maintains exceptionally high directional stability (similarity $\approx 1.0$) consistently throughout the entire process. This confirms that the learned Deterministic Flow effectively approximates an ideal straight-line transport, providing strong empirical support for the single-step SSDF setting. In sharp contrast, the standard FM baseline in Figure S2 (b) exhibits a significant drop in cosine similarity as $t$ approaches 1.0. This phenomenon indicates that the standard FM trajectory undergoes a severe curvature or "hook" near the target distribution. Such non-linear behavior near the endpoint necessitates fine-grained integration steps to accurately resolve the target, explaining why single-step

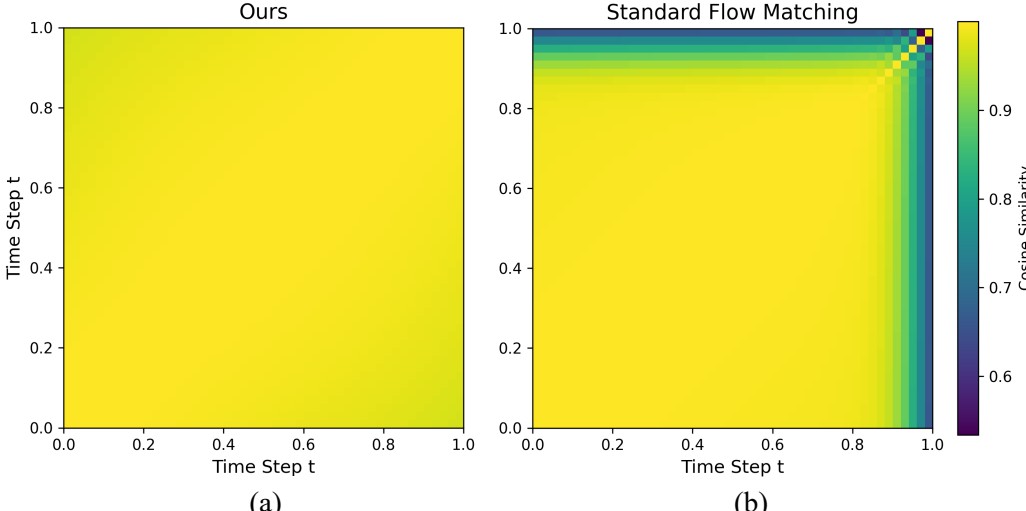

*Figure S2.* **Visual comparison of flow trajectory linearity.** We visualize the directional consistency of the velocity field. (a) Our method maintains consistent velocity direction across time steps, confirming a straight-line path that justifies the use of single-step inference (SSDF). (b) Standard FM shows severe directional variance near $t = 1.0$, implying a curved trajectory at the end of the transport.

inference fails for standard FM.

## E. Object Detection Results

To demonstrate the practical utility of SHADE beyond perceptual metrics, we assess its performance as a data synthesizer for downstream high-level vision tasks. The core objective is to determine if our **flow-based reconstruction** offers a more effective initialization for RAW-domain detectors compared to standard RGB pretraining or competing reconstruction methods.

### E.1. Experimental Setup

We adhere to the evaluation pipeline standardized by prior arts (Berdan et al., 2025), adopting a Pretraining-then-Finetuning paradigm to measure domain transfer capability. The assessment covers two distinct illumination conditions.

For daylight scenarios, we leverage the Cityscapes (Cordts et al., 2016) dataset to synthesize pseudo-RAW data for pretraining. Domain adaptation is then performed by finetuning the model on 300 real RAW images from the PASCALRAW (Omid-Zohoor et al., 2015) training set, with final evaluation conducted on 100 random samples from its test set.

To stress-test robustness in low-light environments, we employ BDD100K (Yu et al., 2020) for pretraining. The model is subsequently finetuned and tested on the NOD (Morawski et al., 2022) dataset (Nikon subset), which is characterized by extreme dynamic range challenges and sensor noise.

The detection framework is built upon YOLOv8. We incorporate data augmentations such as horizontal flipping, random photometric jittering, and noise injection to mitigate overfitting. The pretraining phase spans 50 epochs with a learning rate of $2 \times 10^{-3}$ (batch size 8), followed by a 30-epoch finetuning stage where the learning rate is decayed to $5 \times 10^{-4}$ (batch size 4). Evaluation metrics include mAP, mAP@50, and mAP@75.

### E.2. Performance Analysis

**Daylight Scenarios.** The quantitative evidence for daylight object detection is presented in Table S3. The baseline model, pretrained on standard RGB images from Cityscapes, achieves a mean Average Precision (mAP) of 41.78 when finetuned on real RAW data. This relatively low score reflects the inherent domain shift between the ISP-processed RGB domain and the sensor-specific RAW domain. Existing reconstruction methods partially bridge this gap; for instance, CycleR2R and ReRAW-S improve the performance to 47.75 mAP and 46.90 mAP, respectively. Our method, however, yields a more

*Table S3.* Object detection performance comparison on daylight scenarios.

| Pretraining | Finetuning | mAP ↑ | mAP@50 ↑ | mAP@75 ↑ |
|---|---|---|---|---|
| CS-RGB | GT-RGB | 44.57 | 67.27 | 55.66 |
| CS-RGB | GT-RAW | 41.78 | 67.30 | 47.93 |
| CS-RAW (UPI) | GT-RAW | 46.14 | 69.44 | 57.15 |
| CS-RAW (CycleR2R) | GT-RAW | 47.75 | 72.17 | **58.15** |
| CS-RAW (ReRAW-R) | GT-RAW | 46.71 | 69.59 | 55.35 |
| CS-RAW (ReRAW-S) | GT-RAW | 46.90 | 69.73 | 53.48 |
| **CS-RAW (Ours)** | **GT-RAW** | **50.54** | **74.53** | 58.12 |

*Table S4.* Object detection performance comparison on nighttime scenarios.

| Pretraining | Finetuning | mAP ↑ | mAP@50 ↑ | mAP@75 ↑ |
|---|---|---|---|---|
| BDD-RGB | GT-RGB | 35.40 | 57.18 | 37.20 |
| BDD-RGB | GT-RAW | 33.76 | 55.96 | 33.77 |
| BDD-RAW (UPI) | GT-RAW | 34.46 | 55.79 | 34.68 |
| BDD-RAW (CycleR2R) | GT-RAW | 33.60 | 56.35 | 34.27 |
| BDD-RAW (ReRAW-R) | GT-RAW | 35.50 | 54.21 | 39.53 |
| BDD-RAW (ReRAW-S) | GT-RAW | 35.26 | 56.00 | 34.12 |
| **BDD-RAW (Ours)** | **GT-RAW** | **38.13** | **57.57** | **41.74** |

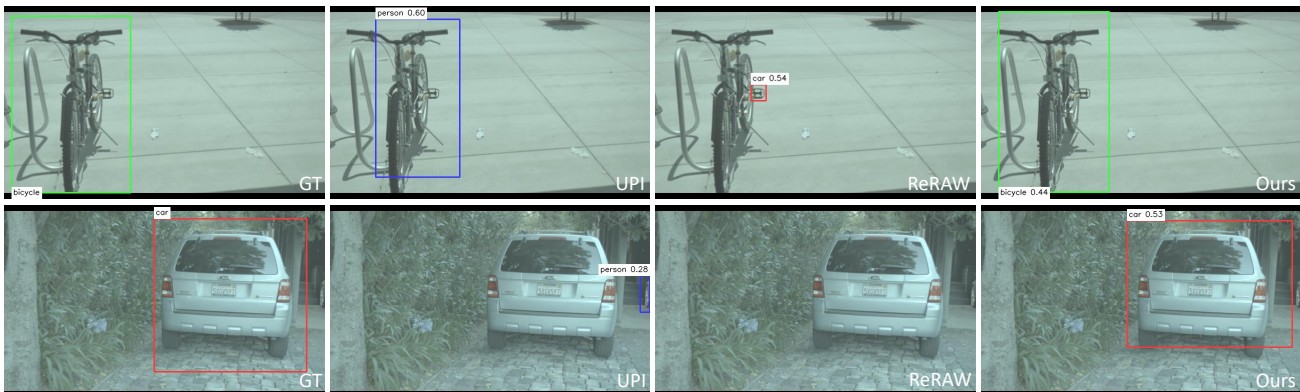

*Figure S3.* Qualitative object detection comparison on daylight scenarios. Competing methods (e.g., UPI, ReRAW) exhibit classification errors, such as misidentifying a bicycle as a person. In contrast, the model trained on our SHADE-synthesized data correctly classifies and localizes objects, matching the Ground Truth (GT).

significant performance gain. By using SHADE-synthesized RAW images for pretraining, the detector achieves 50.54 mAP. This represents a net improvement of +8.76 mAP over the RGB baseline and a margin of +2.79 mAP over the previous best-performing method (CycleR2R). These results suggest that the RAW data generated by our flow-based model provides a closer statistical match to real-world sensor distributions, thereby offering a more effective initialization for the detector.

**Nighttime Scenarios.** Table S4 details the results on the NOD dataset, where low-light conditions introduce complex noise patterns and signal degradation. The RGB pretraining baseline struggles in this regime, recording 33.76 mAP. While competing methods like ReRAW-R provide a lift to 35.50 mAP, our approach demonstrates a stronger capability to handle such degradations, achieving 38.13 mAP. Notably, in terms of high-precision localization (mAP@75), our method scores 41.74, substantially outperforming the RGB baseline (33.77) and ReRAW-R (39.53). The consistent improvements across all metrics indicate that our reconstruction preserves critical semantic information even in regions affected by severe photon noise and underexposure, facilitating better feature learning for downstream detection tasks.

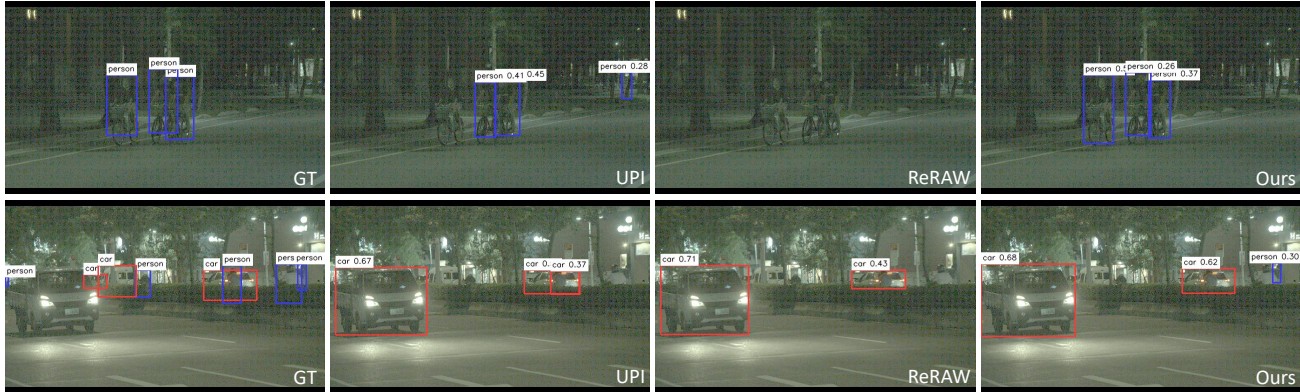

*Figure S4.* Qualitative object detection comparison on nighttime scenarios. In low-light conditions, baseline methods suffer from missed detections (false negatives) due to noise and low contrast. Our method enables the detector to recover obscured objects (e.g., the car in the center of the bottom row) by preserving valid physical priors.

### E.3. Qualitative Analysis of Object Detection

To complement the quantitative metrics, we provide visual comparisons of detection results trained on RAW data synthesized by different methods. Figure S3 and Figure S4 illustrate the qualitative performance in daylight and nighttime scenarios, respectively.

**Daylight Scenarios.** As shown in Figure S3, models trained on data from competing reconstruction methods often struggle with semantic consistency. In the top row, the model trained on UPI-synthesized data incorrectly classifies the bicycle as a "person" (blue box), while ReRAW fails to form a confident bounding box around the object. This suggests that the structural artifacts introduced by these methods corrupt the edge features necessary for fine-grained classification. In contrast, the model trained on our SHADE-synthesized data correctly identifies the object as a "bicycle" (green box) with a bounding box that aligns closely with the Ground Truth (GT), indicating that our method preserves the semantic integrity of the object structure. Similarly, in the bottom row, while competitors fail to detect the vehicle or produce false positives in the background, our method successfully facilitates the detection of the car with accurate localization.

**Nighttime Scenarios.** Figure S4 presents results under challenging low-light conditions, where signal-to-noise ratio is low. In the top row, which features pedestrians and cyclists in a dimly lit street, ReRAW suffers from severe false negatives, missing nearly all targets. UPI detects some instances but yields loose bounding boxes. Our method, benefiting from the flow-based noise modeling, enables the detector to recover these low-contrast objects, producing detection results that are highly consistent with the GT. In the bottom row, the central vehicle is occluded by bushes. While baseline methods miss this object entirely due to the loss of texture details in shadow regions, the model trained on our data successfully recalls the vehicle. This demonstrates that our reconstruction effectively recovers valid feature representations even in regions with signal clipping and heavy noise.

## F. More Qualitative Results

To provide a comprehensive visual assessment of the proposed SHADE framework, we present extensive qualitative comparisons on four benchmark datasets. For each sample, we display the reconstructed RAW image alongside its spatial error map relative to the ground truth, where darker regions indicate lower reconstruction errors and higher fidelity.

### F.1. Daylight Scenarios

Figures S5, S6, and S7 visualize the reconstruction results on the PASCALRAW (Omid-Zohoor et al., 2015), FiveK-Nikon, and FiveK-Canon (Bychkovsky et al., 2011) datasets, respectively. In these scenarios, we compare our method against a comprehensive set of state-of-the-art baselines, including UPI (Brooks et al., 2019), CycleR2R (Li et al., 2024), InvISP (Xing et al., 2021), ReRAW (Berdan et al., 2025), DDPM (Ho et al., 2020), RAW-Diff (Reinders et al., 2025), and RAW-Flow (Liu et al., 2026). As evidenced by the error maps, competing methods often exhibit noticeable structural deviations or color inconsistencies in high-frequency regions. In contrast, SHADE consistently achieves the lowest error magnitude, preserving

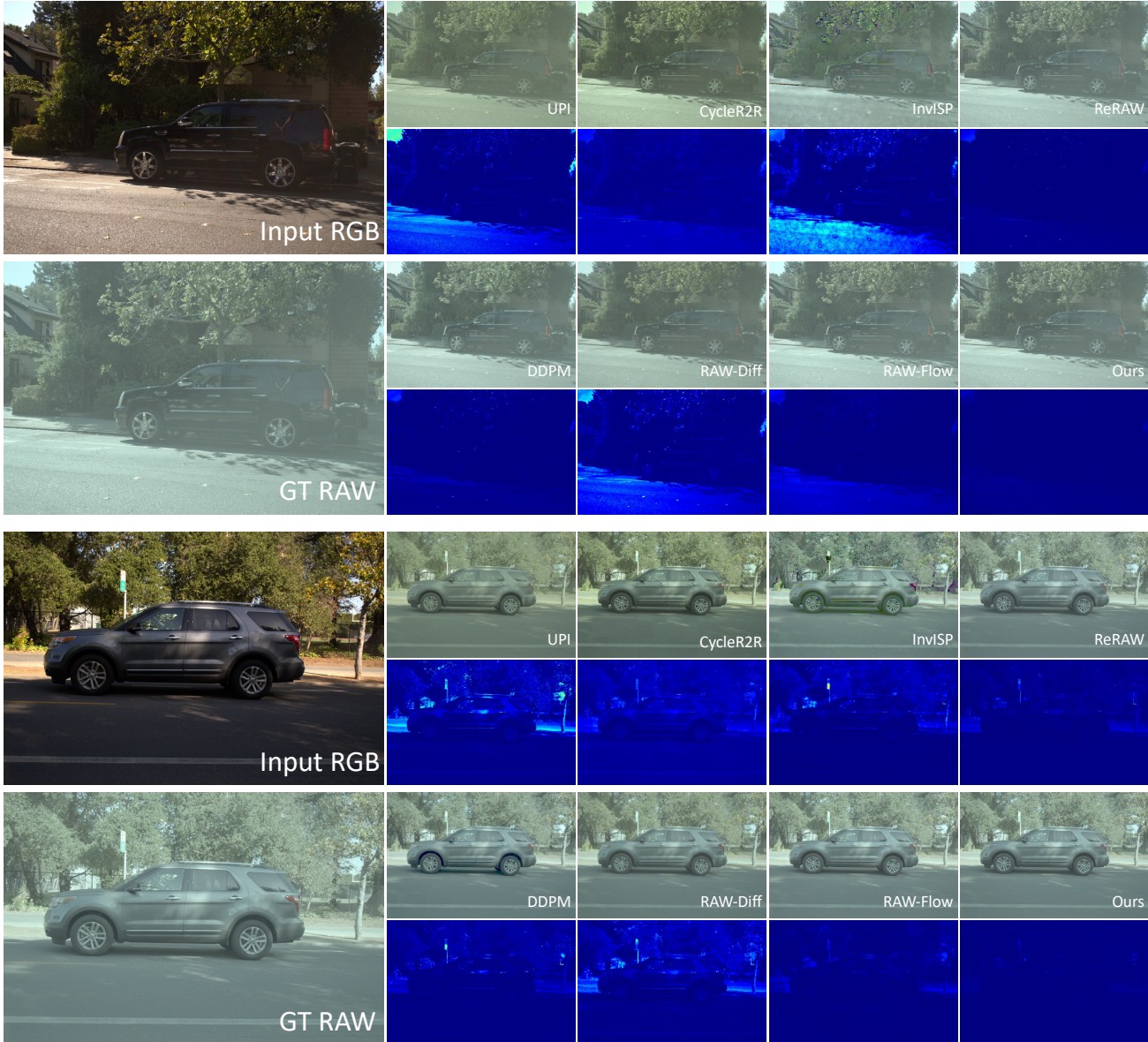

*Figure S5.* More qualitative comparison between our method and other state-of-the-art RGB-to-RAW reconstruction approaches on the PASCALRAW dataset. For each method, we show the reconstructed RAW image and its corresponding error map with respect to the ground-truth RAW image (darker regions indicate lower errors)

sharp details and accurate color fidelity across diverse daylight conditions.

### F.2. Nighttime Scenarios

Figure S8 showcases the performance on the challenging NOD (Morawski et al., 2022) dataset (covering both Sony and Nikon subsets). For these low-light environments, we compare against UPI (Brooks et al., 2019), InvISP (Xing et al., 2021), and ReRAW (Berdan et al., 2025). The results demonstrate that while baseline methods struggle with severe noise and signal clipping inherent to nighttime photography, our method effectively recovers clean and plausible structural details, maintaining stability where other approaches tend to produce artifacts or over-smoothed outputs.

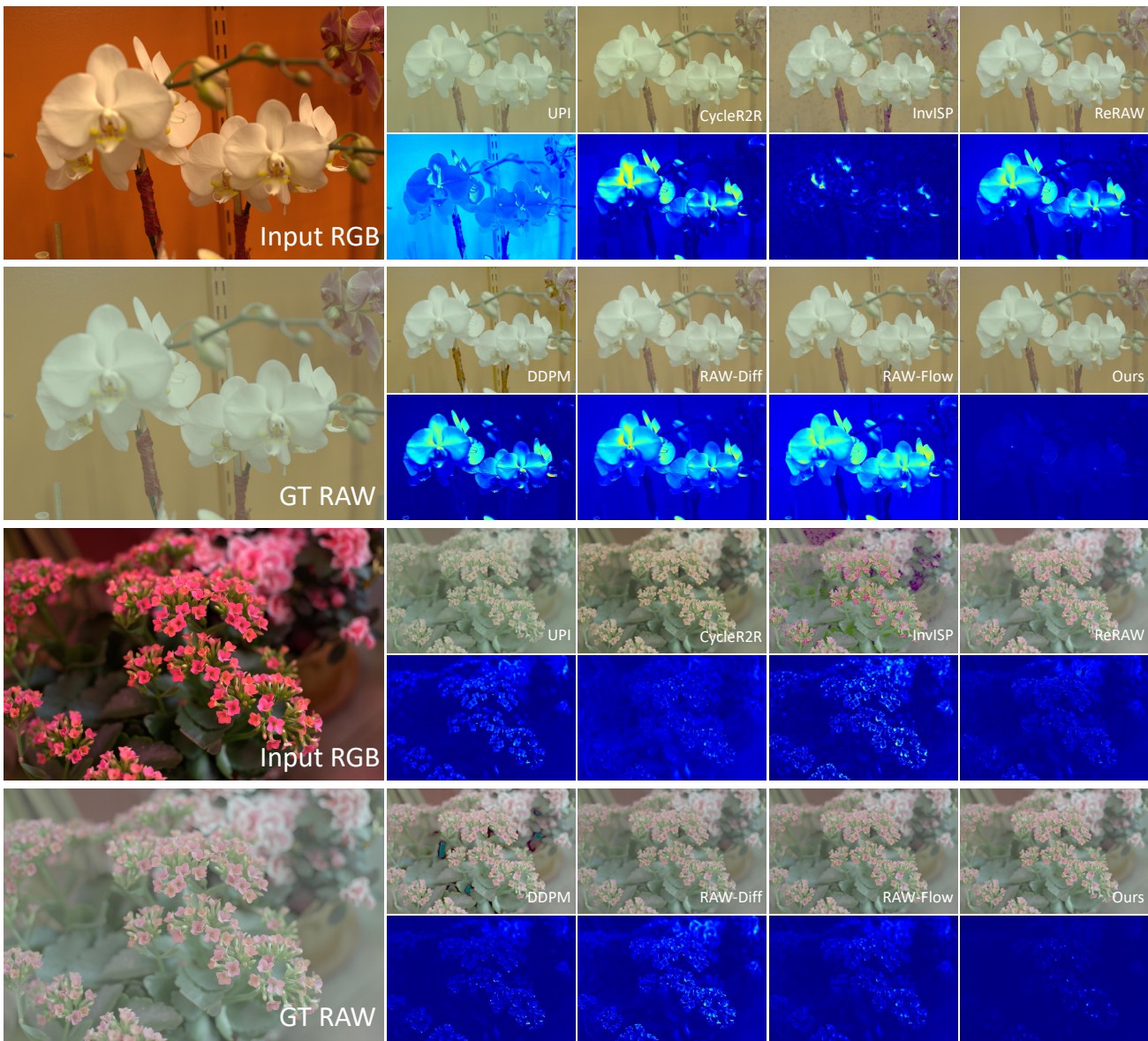

*Figure S6.* More qualitative comparison between our method and other state-of-the-art RGB-to-RAW reconstruction approaches on the Fivek-Nikon dataset. For each method, we show the reconstructed RAW image and its corresponding error map with respect to the ground-truth RAW image (darker regions indicate lower errors)

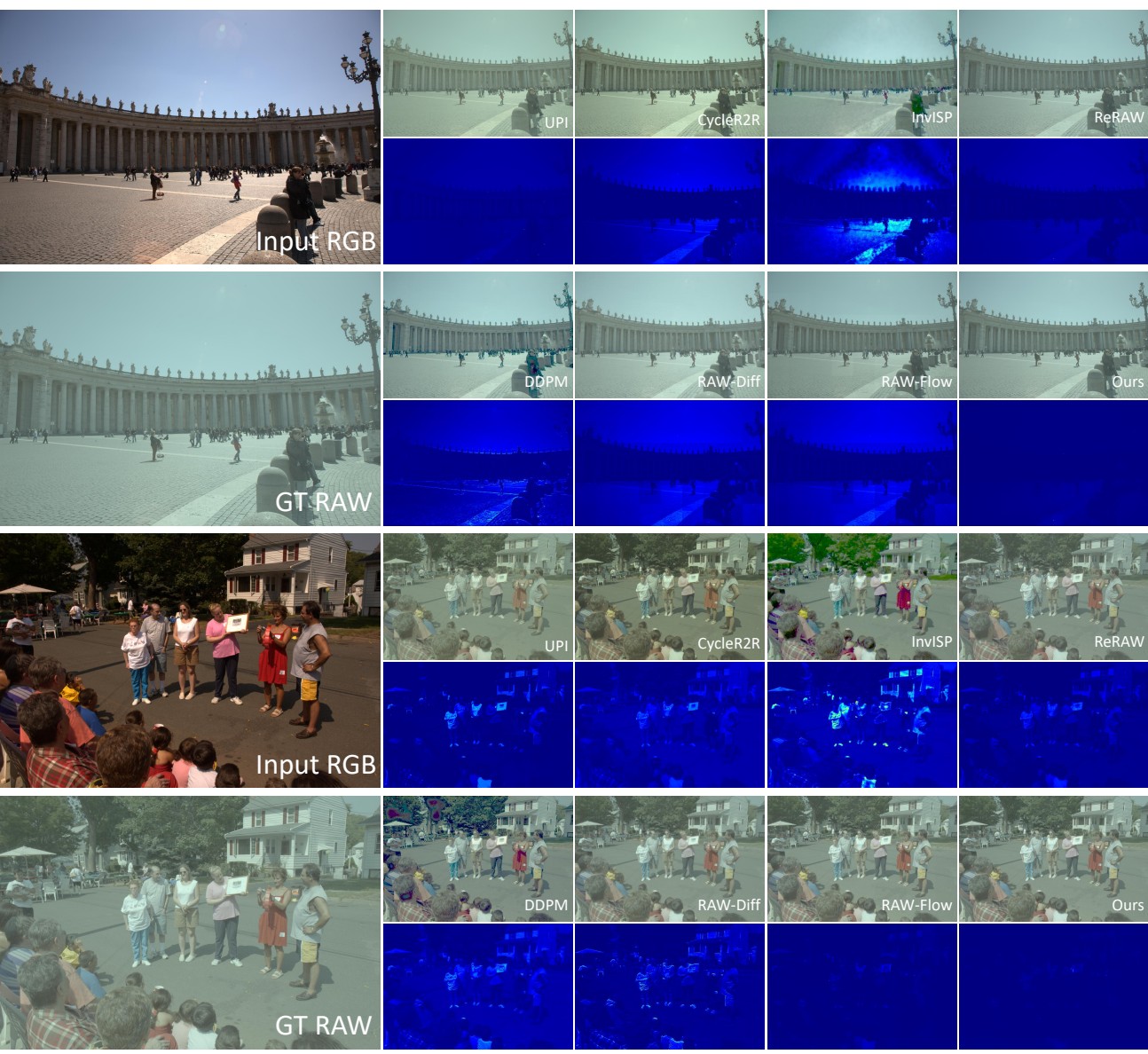

*Figure S7.* More qualitative comparison between our method and other state-of-the-art RGB-to-RAW reconstruction approaches on the Fivek-Canon dataset. For each method, we show the reconstructed RAW image and its corresponding error map with respect to the ground-truth RAW image (darker regions indicate lower errors)

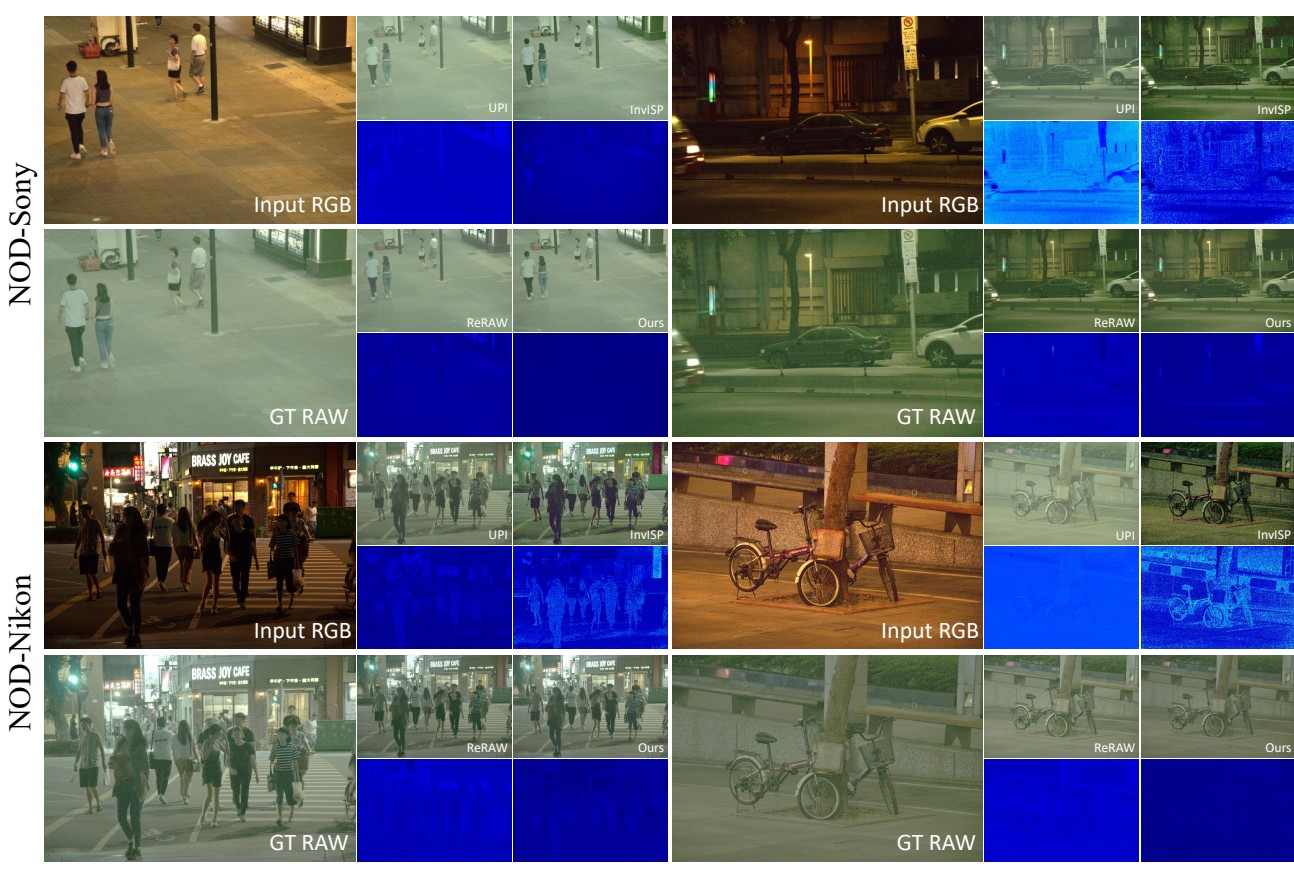

*Figure S8.* More qualitative comparison between our method and other state-of-the-art RGB-to-RAW reconstruction approaches on the NOD-Sony and NOD-Nikon dataset. For each method, we show the reconstructed RAW image and its corresponding error map with respect to the ground-truth RAW image (darker regions indicate lower errors)

