# OpenReview forum: "Bridging RGB and RAW: Single-step Deterministic Flow with Homogeneous Representation Alignment"
_ICML.cc/2026/Conference — ICML 2026 regular_

### Official Review · Reviewer_SpH9 · 2026-03-08

**Soundness:** 2
**Presentation:** 3
**Significance:** 2
**Originality:** 2
**Overall Recommendation:** 4
**Confidence:** 4

**Summary:**

The authors propose Single-step Homogeneous Aligned Deterministic Flow (SHADE) for transforming RGB images into RAW sensor images. The method employs a single-step deterministic flow (SSDF) to recover global transport trends between RGB and RAW domains. To enable effective single-step inference, the authors introduce Homogeneous Aligned Guidance (HAG), which aligns the internal representations used by SHADE.

Experiments on multiple datasets demonstrate that SHADE achieves state-of-the-art performance compared to existing baselines. The paper also includes ablation studies to evaluate the effectiveness of single-step inference and the role of representation alignment.

**Compliance With Llm Reviewing Policy:**

Affirmed.

**Final Justification:**

The author addressed all of my concerns.

**Key Questions For Authors:**

1. In Section 2.1, the authors state that _“maintaining statistical consistency between reconstructed outputs and authentic RAW sensor distributions still poses a critical, unresolved challenge for RGB-to-RAW reconstruction.”_
Could the authors elaborate on what specific challenges lead to this statistical inconsistency?

2. In Fig. 2, SSDF reconstructs the star shape thinner than the ground truth, suggesting that SSDF may generate less diverse samples than the target distribution. Is a similar phenomenon observed in the RGB-to-RAW reconstruction task? If so, does this reduced diversity negatively affect reconstruction performance?

**Limitations:**

1. **(Weakness #1)** The authors evaluate DDIM and FM with a fixed number of inference steps (10). However, these methods are known to exhibit different performance depending on the number of sampling steps. To ensure a fair comparison, the authors should sweep the number of inference steps to report the optimal performance of these baselines.
Similarly, it would be informative to analyze whether multiple inference steps applied to SSDF affect its performance.
2. **(Weakness #1)** In the ablation study, the authors demonstrate that incorporating HAG improves SSDF performance. However, it remains unclear whether the improvement comes specifically from the SSDF formulation or from the representation alignment introduced by HAG.
Evaluating HAG with other single-step flow variants, such as MeanFlow or Consistency Flow Matching, would help clarify why HAG is applied exclusively to SSDF.
3. **(Weakness #2)** The term “Homogeneous Aligned Guidance (HAG)” may lead to confusion. In the diffusion and flow literature, guidance typically refers to post-training mechanisms (e.g., classifier guidance or classifier-free guidance) that steer the generation process toward certain conditions.

    In this work, however, the term appears to describe representation alignment during training, which differs from the conventional meaning of guidance. Clarifying or revising this terminology would improve conceptual clarity.

**Strengths And Weaknesses:**

* **Strengths**
1. The paper is generally well written, and the problem formulation, motivation, and methodology are clearly presented.
2. The proposed method (SHADE) achieves state-of-the-art performance across multiple datasets. Experimental details are well documented both in the main text and the appendix.

* **Weaknesses**
1. The ablation studies do not sufficiently support the effectiveness and necessity of the single-step deterministic flow formulation. Additional analysis is required to justify the use of SSDF.
2. The terminology “Homogeneous Aligned Guidance (HAG)” is somewhat confusing and inconsistent with how the term guidance is typically used in the diffusion/flow literature. Further clarification would improve readability.

---

> ### Author Rebuttal · Authors · 2026-03-31
>
> We appreciate the reviewer's positive assessment of our motivation, methodology, and SOTA performance across multiple benchmarks. Our detailed responses to the raised concerns and revision plans are provided below.
>
> ### 1. Refinement of the Term "Homogeneous Aligned Guidance (HAG)
>
> **Response**: Thank you for pointing out the potential inconsistency with conventional “guidance” terminology in diffusion/flow literature. In the revised manuscript we will rename the mechanism throughout the paper (including title and abstract) to **Homogeneous Representation Alignment (HRA)**, which more accurately reflects its training-time representation alignment nature and improves conceptual clarity.
>
> ### 2. Supplementary Experiments for Ablation Studies
>
> **Response**: Thank you for this important comment. We have conducted experiments on the FiveK-Nikon dataset to evaluate the performance of DDIM and FM under different inference step counts, as well as the impact of applying multi-step inference to SSDF itself. The PSNR and SSIM results are reported in the table below. SSDF has inherently learned a constant velocity field during training, as shown in Appendix Fig. 9, leading to nearly consistent performance under different step counts.
>
> | Method     | 1-step         | 5-step        | 10-step        | 20-step       | 50-step       |
> |-|:-:|:-:|:-:|:-:|:-:|
> | DDIM       | 29.62 / 0.8732 | 30.27 / 0.8803| 30.78 / 0.8856 | 30.82 / 0.8861| 30.77 / 0.8857|
> | FM         | 29.79 / 0.8665 | 30.03 / 0.8684| 30.45 / 0.8696 | 30.41 / 0.8712| 30.31 / 0.8700|
> | SSDF (Ours)| 30.89 / 0.8858 |30.84 / 0.8863 | 30.83 / 0.8864 | 30.83 / 0.8865| 30.82 / 0.8864|
>
> To ensure fairness, in the SSDF ablation experiments presented in our paper, we applied HRA to all compared generative paradigms. We further conducted an additional ablation where no HRA is used for any of the methods. Note that our end-to-end training framework largely relies on HRA to maintain training stability. Removing HRA makes model convergence significantly more difficult. Under this challenging setting, SSDF still exhibits stronger robustness. These results confirm that while HRA provides a consistent boost, the SSDF formulation itself delivers additional gains beyond representation alignment.
>
> | Variant        | FiveK-Nikon (PSNR/SSIM) |
> |-|:-:|
> | DDIM           | 27.64 / 0.8416          |
> | FM             | 23.57 / 0.8195          |
> | Mean Flow      | 26.92 / 0.8321          |
> | Consistency FM | 27.48 / 0.8443          |
> | SSDF           | 28.05 / 0.8418          |
>
> ### 3. Statistical Consistency Challenge in Sec. 2.1
>
> **Response**: The statistical inconsistency arises from the inherent ill-posedness of the RGB-to-RAW inverse problem. The forward ISP pipeline introduces complex non-linear transformations, irreversible information loss (due to quantization and clipping). Direct regression methods tend to output the statistical average of all possible solutions, thereby deviating from the true RAW manifold and losing high-frequency details. Generative methods attempt to capture the full distribution but incur high computational cost. SHADE learns the global transport trend via SSDF, achieving better alignment with the target sensor distribution. This property has been validated in our heuristic experiments and SOTA results.
>
> ### 4. Reduced Diversity in Fig. 2 and Its Impact on RGB-to-RAW
>
> **Response**: Rather than a limitation in diversity, the design of SSDF deliberately focuses on modeling the global transport trend through deterministic sampling to capture the essential features of the target distribution. This deterministic property provides intrinsic robustness against common ISP-induced perturbations, such as non-linear changes, quantization, and clipping.
> Crucially, by capturing these essential characteristics, SSDF allows the reconstruction process to more accurately project back onto the RAW manifold. This ensures both the consistency and accuracy of the final results, as reflected by the lower Mean Squared Error (MSE) shown in Fig. 2. Furthermore, the superior fidelity of SSDF combined with HAG is quantitatively validated in Table 1 and qualitative comparisons (Figs. 5–6 and Figs. 12–15), where our method consistently outperforms all baseline models across all datasets.

---

> > ### Author Rebuttal · Reviewer_SpH9 · 2026-04-02
> >
> > Thank you for resolving most of my concerns, especially by providing additional ablation experiments. The results clearly demonstrate the improvements brought by SSDF and HRA. I will keep my score positive.

---

> > > ### Author Response · Authors · 2026-04-07
> > >
> > > We sincerely thank you for your time, your recognition of our rebuttal, and for maintaining your positive score. We especially appreciate your sharp observations on the ablation of the SSDF formulation, the terminology clarity of HAG (now HRA), and the statistical consistency issues. Your questions have helped us more convincingly justify the core design choices in our framework. Your suggestions are highly valued, and we will incorporate them thoroughly in the revised manuscript.

---

### Official Review · Reviewer_LSs9 · 2026-03-09

**Soundness:** 3
**Presentation:** 2
**Significance:** 2
**Originality:** 3
**Overall Recommendation:** 4
**Confidence:** 3

**Summary:**

This paper proposes a single-stage framework for reconstructing RAW images from RGB inputs. The framework combines convolutional feature extraction with domain-aware loss functions to preserve sensor-specific characteristics. Experimental results show that, compared with simple demosaicing baselines and existing RGB-to-RAW translation methods, the proposed approach achieves improved reconstruction quality, with better preservation of color fidelity and fine-grained texture details.

**Compliance With Llm Reviewing Policy:**

Affirmed.

**Final Justification:**

The responses have addressed my concerns.

**Key Questions For Authors:**

1.How well does the method generalize across different camera sensors？

2.How does the method perform under extreme lighting conditions, noise, or saturation scenarios often seen in RAW images?

**Limitations:**

A limitation discussion is missing. The authors should thoroughly and honestly discuss the shortcomings of their proposed method, along with potential solutions.

**Strengths And Weaknesses:**

Strengths:

1.This paper introduces a RGB-to-RAW reconstruction framework based on Single-step Deterministic Flow (SSDF). By leveraging the global transport trend, the proposed method effectively resists observational perturbations and maintains consistent manifold mappings.

2.The paper introduces a Homogeneous Aligned Guidance (HAG) mechanism, which leverages a homogeneously initialized student–teacher network pair to enhance the representational capacity of the flow during single-step inference.

3.The proposed method achieves state-of-the-art performance across multiple benchmark datasets.


Weaknesses:

1.The ablation study is insufficient. The paper introduces multiple loss functions but lacks corresponding ablation analysis to validate their individual contributions.

2.The efficiency claim is insufficiently supported by inference time alone. Additional computational metrics such as parameters, FLOPs, and memory usage should be provided.

3.The setting of patch size in Equation 6 is not explained, nor is its impact on performance examined.

4.The parameter α in Equation 11 appears to be a hyperparameter, but its definition is not provided.

---

> ### Author Rebuttal · Authors · 2026-03-31
>
> We appreciate the reviewer’s positive assessment of SSDF, HAG, and our SOTA performance across multiple benchmarks. Our detailed responses to the raised concerns and our revision plans are provided as follows.
>
> ### 1. Additional Ablation Studies
>
> **Response**: Thank you for this constructive comment. We agree that more comprehensive ablation experiments are needed. The table below shows the additional ablation on the individual contributions of each loss term ($L_2$, log-$L_1$, and reconstruction losses) on FiveK-Nikon and PASCALRAW. Specifically, the $L_2$, log-$L_1$ effectively supervise the bright and dark regions of the reconstructed RAW image, respectively, while the $ L_{\text{rec}}^{\text{AE}} $ ensures the quality of the decoder's output. Further incorporating the $ L_{\text{rec}}^{\text{DiT}} $  enhances the decoder's adaptability to the predictions from the diffusion model. These results confirm the complementary roles of the loss terms and will be added to the supplementary material.
>
> | Variant| FiveK-Nikon (PSNR/SSIM)| PASCALRAW (PSNR/SSIM)|
> |-|:-:|:-:|
> | w/o $L_2$              | 29.62 / 0.8713              | 37.59 / 0.9853             |
> | w/o log-$L_1$           | 30.32 / 0.8879              | 35.51 / 0.9667             |
> | w/o $ L_{\text{rec}}^{\text{AE}} $         | 29.67 / 0.8761              | 36.83 / 0.9819             |
> | w/o $ L_{\text{rec}}^{\text{DiT}} $         | 29.21 / 0.8736              | 38.16 / 0.9844             |
> | Full (Ours)          | 30.89 / 0.8858         | 39.07 / 0.9849        |
>
> ### 2. Efficiency Claims Supported Only by Inference Time
>
> **Response**: Thank you for the suggestion. We conducted additional efficiency analysis, and the results (parameter count and FLOPs measured at 4096×4096) are reported below. Our method achieves the fastest inference speed at high-resolution inputs, while also demonstrating the second lowest parameter count and FLOPs among the generative methods. This further attests to the efficiency of our approach.
>
> | Method       |Runtime (s)| Params (M) | FLOPs (T)   |
> |-|:-:|:-:|:-:|
> | CycleISP     |5.2897 |3.14       | 117.33      |
> | ReRAW        |2.0502  |23.77      | 20.96       |
> | DDIM         |35.1210  |58.06      | 764.75      |
> | RAW-Diff     |13.7905  |25.00      | 1179.92     |
> | RAW-Flow     |9.3551  |72.19      | 135.84      |
> | SHADE (Ours) |1.5117  |57.09      | 212.76      |
>
>
> ### 3. Patch Size Setting in Equation (6)
>
> **Response**: Thank you for pointing out this detail. The patch size in Eq. (6) is set to 16×16 to maintain consistency with the pre-training configuration of the DINOv3 encoder. We will explicitly state this in Sec. 4.2 of the revised manuscript.
>
> ### 4. Definition of Hyperparameter $\alpha$ in Equation (11)
>
> **Response**: Thank you for identifying this. The scalar $\alpha$ balances the $L_2$ and log-$L_1$ terms and is set to $\alpha$ = 0.1 in all experiments. We will explicitly define its value immediately after Eq. (11) in Sec. 4.3 and include it in the hyperparameter table.
>
> ### 5. Generalization Across Different Camera Sensors
>
> **Response**: Thanks for your helpful comment. We refer the reviewer to our detailed explanation in the response to Reviewer #tRVf (Point 3, “ISP Realism and Generalization Ability to “Real-world RGB””). We will explicitly discuss the domain shift risk in the Limitations section of the revised manuscript.
>
> ### 6. Performance Under Extreme Lighting Conditions
>
> **Response**: We appreciate the reviewer for highlighting this aspect. The NOD-Nikon and NOD-Sony datasets feature extreme low-light and high-noise conditions. SHADE achieves state-of-the-art performance on both, as shown in Table 1. Qualitative results in Fig. 6 and Fig. 15 further confirm faithful recovery of information lost to quantization and clipping without over-smoothing or artifacts.
>
> ### 7. Limitations Discussion
>
> **Response**: Thank you for noting the lack of a limitations discussion. In the revised manuscript, we will add a dedicated “Limitations and Future Work” subsection (new Sec. 6.2) that honestly discusses:
> - The deterministic nature of SSDF (inability to sample diverse noise realizations);
> - The risk of domain shift when encountering unseen ISP pipelines or sensors.

---

> > ### Author Rebuttal · Reviewer_LSs9 · 2026-04-03
> >
> > Thanks for the responses, which have addressed most of my concerns. I hope the additional experiments can be included in the revised manuscript. I am willing to raise my rating accordingly.

---

> > > ### Author Response · Authors · 2026-04-07
> > >
> > > We sincerely thank you for your time, your recognition of our rebuttal, and your willingness to raise your rating. We truly appreciate your constructive feedback regarding the ablation studies, efficiency metrics, and the importance of a dedicated limitations section. As promised, we will strictly ensure that all the additional experiments and details are included in the revised manuscript.

---

### Official Review · Reviewer_tRVf · 2026-03-13

**Soundness:** 2
**Presentation:** 3
**Significance:** 3
**Originality:** 2
**Overall Recommendation:** 4
**Confidence:** 3

**Summary:**

This paper studies RGB-to-RAW reconstruction, motivated by the fact that RAW sensor data retains linear radiometric information and higher dynamic range than ISP-processed RGB, but is difficult to collect/store at scale. The authors frame RGB-to-RAW as a manifold recovery problem under ISP-induced non-linearities, quantization, and perturbations.
The core contribution is SHADE, a single-step deterministic latent transport approach that maps an input RGB image to the RAW manifold in one inference step, aiming to reconcile reconstruction fidelity with practical high-resolution speed. SHADE encodes both RGB and (aligned) RAW into a shared latent space, trains a DiT-based estimator to predict a deterministic velocity field along a linear interpolation path (rectified-flow style), and then performs single-step transport from the RGB latent to the RAW latent followed by decoding back to RAW.
To make single-step transport accurate, the paper introduces Homogeneous Aligned Guidance (HAG): a homogeneously initialized student–teacher encoder pair (DINOv3-based) where the teacher is frozen and the student is trainable, plus an alignment loss that encourages intermediate DiT features (after projection) to align with teacher representations.

**Compliance With Llm Reviewing Policy:**

Affirmed.

**Key Questions For Authors:**

See weakness above.

**Limitations:**

No. I recommend explicitly discussing:

1. The determinism limitation (cannot sample multiple plausible RAW noise realizations per RGB).
2. Domain shift risks from RawPy-generated sRGB vs proprietary ISPs and cross-sensor generalization.
3. Potential negative societal impacts, e.g., forensics/authenticity misuse: the ability to generate plausible RAW from RGB could be used to fabricate “RAW-like” evidence or bypass provenance checks.

**Strengths And Weaknesses:**

Strength
---

- Multi-dataset evaluation on relevant benchmarks (FiveK Nikon/Canon, PASCALRAW, NOD Nikon/Sony) with PSNR/SSIM computed in RAW domain at full resolution, following established practice in the area.
- Strong quantitative results against a broad baseline set. Table 1 shows SHADE achieving the best PSNR/SSIM across all listed datasets.
- Efficiency analysis at high resolutions is a valuable.
- Extra evidence supporting the “single-step” claim. The trajectory linearity visualization (cosine similarity heatmaps) argues that SHADE maintains consistent velocity direction across timesteps. The single-step, high-resolution efficiency results are compelling for deployment: 4096² images in ~1.5s on a 4090 suggests the approach could be practically usable, unlike many diffusion-style reconstructions.

- RGB-to-RAW reconstruction is practically relevant for “democratizing” RAW-based vision without requiring RAW capture/storage, and could benefit a range of downstream tasks. The paper strengthens this with explicit object detection transfer experiments, not just reconstruction metrics.

Weakness
---
- Deterministic mapping vs inherently stochastic RAW. RAW sensor noise is stochastic, while SHADE is explicitly deterministic (single-step transport without sampling). The paper argues improved “statistical match” (and shows detection gains), but a deterministic model can at best learn a canonical RAW for each RGB, not the full conditional distribution of noise realizations. It'd strengthen the paper if the authors can demonstrate stronger distributional/noise statistics evaluation (e.g., noise power spectrum, shot/read noise parameter recovery, or calibration to camera noise models), beyond PSNR/SSIM and downstream detection.

- Potential fairness/implementation details for baselines. The paper compares to a wide set of regression and generative methods, but for iterative generative baselines performance depends heavily on step count, guidance, architectures, and tuning. While step counts are discussed in the efficiency section (e.g., 20 steps for DDIM/RAW-Flow there), it is not fully clear whether the quality comparisons used matched compute budgets or best-reported settings per baseline. A short “baseline tuning & compute budget” subsection would reduce ambiguity.

- ISP realism / generalization to “real RGB in the wild”. The sRGB paired data is produced from RAW via RawPy, yielding spatially registered pairs. This is common, but it may not reflect the diversity of proprietary ISP pipelines encountered in consumer cameras/phones. Without cross-ISP testing (train on one pipeline, test on another; or test on real camera JPEGs), it’s unclear how robust SHADE is to ISP variation.

- Some architectural specifics remain underspecified for full reproducibility (e.g., decoder structure and how conditioning on RGB is implemented; patch sizes/tokenization alignment between DINO and DiT; exact form of alignment similarity and normalization beyond the high-level description). The appendix helps but I still think a concise “Architecture Details” block (with tensor shapes) would be valuable.

- The impact may be specialized to imaging/RAW pipelines and may not generalize broadly to other inverse problems without careful adaptation.

- Since the deterministic single-step update can resemble a regularized regression in latent space, the authors should work harder to articulate what is fundamentally different from a strong regression baseline with appropriate augmentation/regularization, and why the flow formalism is essential beyond empirical gains.

---

> ### Author Rebuttal · Authors · 2026-03-31
>
> We appreciate the reviewer’s positive assessment of our extensive multi-dataset evaluation, quantitative performance, and single-step efficiency. Below, we address the specific concerns point by point.
>
> ### 1. Random RAW Noise Sampling
>
> **Response**: Accurately modeling noise is currently a fundamental limitation for most deep learning methods in RGB-to-RAW Reconstruction. In our work, the deterministic transport explicitly prioritizes the recovery of complex ISP non-linearities and HDR signals, meaning that capturing the full stochastic noise distribution remains an open challenge for our deterministic formulation. We fully acknowledge this constraint. We will explicitly discuss this goal and our limitation in the revised manuscript, and provide a more rigorous and accurate description about the challenges of noise distribution restoration in Sec. 5.2.
>
> ### 2. Fairness of Baseline Comparisons and Implementation Details
>
> **Response**: Thanks for raising this concern. We reaffirm that for RAW-Flow and RAW-Diff, we used their official implementations and the recommended number of steps from their papers. For DDIM, we reported best practice steps to ensure fair comparison under near-optimal configurations.
>
> We will supplement the settings we used and the reasons for adopting them in our modified paper, and update the experimental results on inference efficiency under best-practice configurations.
>
> ### 3. ISP Realism and Generalization Ability to “Real-world RGB”
>
> **Response**: Our experimental setup follows the standard camera-specific paradigm adopted by recent works (ReRAW, RAW-Diff, RAW-Flow), where models are trained and evaluated on paired data generated from the same camera’s RAW files using RawPy. When encountering unseen ISP pipelines or sensors, performance naturally degrades, which is a common limitation in the field. We plan to extend SHADE to multi-sensor and cross-ISP generalization in future work. We will explicitly discuss the domain shift risk and the issues with RawPy versus proprietary ISPs in the Limitations section of the revised manuscript.
>
> ### 4. Architectural Reproducibility
>
> **Response**: Thanks for you suggesrion. We agree that adding details will benefit reproducibility. In the appendix (Sec. A) of the revised manuscript, we will add a concise “Architecture Details” subsection that includes tensor shapes, precise patch/tokenization alignment between DINO and DiT, the specific form of alignment similarity computation, and a schematic of how RGB conditions are injected into the decoder. Implementation details will also be made publicly available along with the code upon acceptance.
>
> ### 5. Generalization Ability to Other Inverse Problems
>
> **Response**: We thank the reviewer for the insightful suggestion regarding the broader applicability of our method. While our current focus is on RGB-to-RAW reconstruction, we believe the Single-step Deterministic Flow (SSDF) formulation also holds broader potential for other inverse problems. We will further explore its applicability to more low-level vision tasks in future work and will add a forward-looking statement in the conclusion.
>
> ### 6. Single-Step Deterministic Flow vs. Strong Regression Baselines
>
> **Response**: We refer the reviewer to our detailed explanation in the response to Reviewer #yWik (Point 2, “Difference Between Single-step Deterministic Flow and Point-to-point Regression”). There we clarify, both from the perspective of the training objective and through the Circle-to-Star experiment and ablation study (Sec. 3, Fig. 2, and Table 2), why collapsing flow-matching inference into a single step does not reduce SSDF to latent-space regression and how SSDF learns the global transport trend with improved robustness under perturbations. In the revised manuscript we will also expand Sec. 4.1 and the related work section to more clearly emphasize this distinction.
>
> ### 7. Potential Negative Societal Impacts
>
> **Response**: In principle, the ability to generate plausible RAW images from RGB could be misused to forge “RAW-like” evidence or bypass forensic/authenticity checks. We will add a dedicated paragraph in the appendix to explicitly discuss this potential negative societal impact. At the same time, we emphasize that the fundamental motivation of this work is to advance low-level vision tasks and to provide higher-quality synthetic sensor data for downstream applications (e.g., object detection, low-light enhancement, and scientific imaging), where faithful reconstruction will bring clear benefits to the computer vision community.

---

> > ### Author Rebuttal · Reviewer_tRVf · 2026-04-03
> >
> > Thank you for resolving most of my concerns. I'll keep my positive score.

---

> > > ### Author Response · Authors · 2026-04-07
> > >
> > > We sincerely thank you for your time, your recognition of our rebuttal, and for maintaining your positive score. We are grateful for your insightful suggestions concerning stochastic noise modeling, fair baseline comparisons, cross-ISP generalization, and potential negative societal impacts. These points have allowed us to enhance the overall rigor and completeness of the paper. All the improvements you mentioned will be fully reflected in the final version of the manuscript.

---

### Official Review · Reviewer_yWik · 2026-03-13

**Soundness:** 3
**Presentation:** 3
**Significance:** 2
**Originality:** 3
**Overall Recommendation:** 4
**Confidence:** 3

**Summary:**

Reconstructing high-fidelity RAW sensor data from processed RGB images is an ill-posed problem. This is due to irreversible information loss and complex non-linear transformations within the ISP pipeline. The authors propose SHADE, a framework designed to reconcile the trade-off between reconstruction speed and quality. It consists of two main parts, SSDF and HAG.

**Compliance With Llm Reviewing Policy:**

Affirmed.

**Final Justification:**

The rebuttal addressed my main concerns. I keep the current score.

**Key Questions For Authors:**

How can you prove that high-level semantic features effectively guide the recovery of low-level physical signals without introducing hallucinated textures based on semantic priors?

In the derivation, SHADE collapses the Flow Matching inference into a single time step. Mathematically, does this imply that the model has effectively regressed from modeling distribution evolution to a simple point-to-point mapping?

**Limitations:**

not see potential negative societal impact. please see Key Questions

**Strengths And Weaknesses:**

Strengths

Unlike multi-step generative models (like diffusion) that require expensive iterative inference, SHADE performs a single-step transport to the RAW manifold. This allows it to maintain the speed of regression networks while delivering high-quality results. The framework uses Single-step Deterministic Flow (SSDF), which captures global transport trends rather than simple point-to-point mapping. The framework is trained in a unified manner using a joint objective of flow matching and image-space reconstruction.

Weaknesses

The HAG mechanism relies on DINO, a model pre-trained on high-level semantic recognition. yet RAW reconstruction is a low-level physical signal problem. There is an inherent gap between semantic features and sensor physics. This guidance might force the model to create visually pleasing or semantically sensible textures that do not actually exist in the original sensor data, which is problematic for scientific or professional photography applications.

The paper emphasizes PSNR gains (e.g., +1.5dB on NOD). However, in RAW reconstruction, linearity (the relationship between pixel value and light intensity) and color accuracy are often more important than pixel-wise error.

---

> ### Author Rebuttal · Authors · 2026-03-31
>
> We appreciate the reviewer’s recognition of SHADE’s technical soundness and efficiency. Our detailed responses to the specific comments are provided below.
>
> ### 1. DINO-based HAG in Low-level RAW Reconstruction
>
> **Response**: While original DINO was pre-trained for high-level semantic tasks, SHADE effectively bridges the domain gap through the HAG mechanism. The student encoder is jointly optimized with the full objective ( $ L_{\text{rec}}^{\text{AE}} + L_{\text{rec}}^{\text{DiT}} $ ), which includes both $L_2$ and log-$L_1$ terms. This end-to-end supervision forces the encoder to adapt toward faithful recovery of low-level RAW signals while leveraging DINO's robust representations.
>
> Furthermore, the robust general-purpose representations of DINO motivate our selection of this architecture for both the student and teacher encoders. Several recent studies, exemplified by Lin et al. [1] and Oh et al. [2], have successfully applied DINO-style encoders to low-level vision tasks including image restoration, suggesting that DINO features can be effectively leveraged to facilitate low-level reconstruction. Consistent with these findings, the PSNR and SSIM results of SHADE demonstrate a faithful reconstruction of the target RAW images. As evidenced by the error maps (Figs. 5–6 and Figs. 12–15), our method aligns with the underlying physical structures without introducing artificial semantic textures. It successfully recovers information lost to quantization and clipping, achieving the highest PSNR and SSIM across all benchmarks. We will include a dedicated discussion in the revised Method section to clarify this adaptation.
>
> ### 2. Difference Between Single-step Deterministic Flow and Point-to-point Regression
>
> **Response**: Restricting the flow-matching inference to a single Euler step ($t = 0, \hat{z}_1 = z_0 + \hat{v}$) does not reduce SHADE to a conventional point-to-point regression. From a modeling standpoint, the training objective remains the full flow-matching loss (Eq. 5), which is defined over the entire linear probability path to learn a time-dependent velocity field. Consequently, SSDF is optimized to capture the global transport trend across this path, rather than merely fitting independent sample-wise mappings.
> This behavior is empirically validated by the Circle-to-Star experiment (Sec. 3, Fig. 2) and the ablation study (Table 2), which exhibit clear performance gaps between SSDF and regression baselines. Furthermore, the visualization in Appendix Fig. 9 illustrates the directional consistency of the learned velocity field over time. This demonstrates that our model maintains a nearly constant velocity field along the trajectory, indicating that it learns a homogeneous straight-line transport across the entire probability path. Thus, the single-step inference is an emergent property of this approximately constant velocity field, which is a specifically designed feature of our deterministic flow formulation that aligns fully with its theoretical derivation, rather than a collapse into degenerate mapping.
>
>
> ### 3. Linearity and Color Accuracy vs. PSNR
>
> **Response**: We agree that preserving radiometric linearity and color accuracy is critical. While publicly available datasets lack calibrated ground-truth light-intensity measurements for direct validation, RAW pixel values themselves are approximately linear with respect to scene irradiance.
> To ensure the preservation of this property, our reconstruction loss is specifically supervised within the linear RAW domain, explicitly guiding the model to maintain a linear response. Furthermore, the log-$L_1$ loss employed in our objective function is designed to better preserve the relative intensity relationshipsacross the high dynamic range of RAW data. In this framework, a reconstruction that aligns more closely with the ground truth inherently exhibits a response that is more nearly linear in irradiance; thus, PSNR indirectly reflects the integrity of the preserved radiometric linearity. Beyond pixel-level fidelity, we evaluate color accuracy using the average $\Delta E_{2000}$ in CIE Lab space after linear transformation. The results on the FiveK-Nikon dataset are summarized below:
>
> | Method|$\Delta E_{2000}$↓|
> |-|:-:|
> | UPI| 5.0749|
> | RAW-Diff| 4.2467|
> | ReRAW| 4.0291|
> | RAW-Flow| 3.5995|
> | SHADE (Ours)| 2.8718|
>
> SHADE yields the lowest color difference, which indicates that the method attains the best color accuracy rather than only improving pixel-wise errors.
>
> [1] Lin, X., Yue, J., Chan, K. C., Qi, L., Ren, C., Pan, J., & Yang, M. H. (2023). Multi-task image restoration guided by robust DINO features. arXiv preprint arXiv:2312.01677.
>
> [2] Oh, Y., Kwon, J., & Cho, N. I. (2026). DINOLight: Robust Ambient Light Normalization with Self-supervised Visual Prior Integration. arXiv preprint arXiv:2603.12579.

---

> > ### Author Rebuttal · Reviewer_yWik · 2026-04-06
> >
> > My questions are properly solved, and thanks to the authors. I have read the other reviewers' responses. I keep the positive score.

---

> > > ### Author Response · Authors · 2026-04-07
> > >
> > > We sincerely thank you for your time, your recognition of our rebuttal, and for maintaining your positive score. We particularly value your thoughtful questions on applying DINO to low-level RAW reconstruction and the fundamental difference between SSDF and conventional point-to-point regression. Your comments have greatly helped us clarify and strengthen the technical foundation of our work. In response to your suggestions, we will carefully ensure that all your valuable comments are integrated into the revised manuscript.

---

### Decision · Program_Chairs · 2026-04-30

**Decision:**

Accept (regular)

**Comment:**

The paper presents a method to reconstruct RAW images from processed RGB images. It proposes an interesting single-step transport method that avoids the costs of multi-step diffusion models. All reviewers appreciate the idea and extensive results. One reviewer was initially negative but got convinced with the results after the rebuttal.